# Kiki or Bouba? Sound Symbolism in Vision-and-Language Models

**Morris Alper and Hadar Averbuch-Elor**
Tel Aviv University

## Abstract

Although the mapping between sound and meaning in human language is assumed to be largely arbitrary, research in cognitive science has shown that there are non-trivial correlations between particular sounds and meanings across languages and demographic groups, a phenomenon known as *sound symbolism*. Among the many dimensions of meaning, sound symbolism is particularly salient and well-demonstrated with regards to cross-modal associations between language and the visual domain. In this work, we address the question of whether sound symbolism is reflected in vision-and-language models such as CLIP and Stable Diffusion. Using zero-shot knowledge probing to investigate the inherent knowledge of these models, we find strong evidence that they do show this pattern, paralleling the well-known *kiki–bouba effect* in psycholinguistics. Our work provides a novel method for demonstrating sound symbolism and understanding its nature using computational tools. Our code will be made publicly available[1].

## 1 Introduction

*"What's in a name? That which we call a rose by any other name would smell as sweet."*

> —*William Shakespeare, Romeo and Juliet*

Philosophers of language have long debated whether the mapping between sound and meaning in speech is arbitrary. Discussions on this topic date back to the Socratic dialogues of Plato, as well as early modern philosophers such as John Locke and Gottfried Wilhelm Leibniz [30]. Charles de Saussure, the seminal early 20th century linguist and semiotician, famously stated that *le signe est arbitraire*[2]. In Saussure's view, words are simply arbitrary conventions and their sounds have no inherent connection to their meanings; hence the French word *chien* and its English translation *dog* both equally denote the same animal despite sharing no sounds in common [15].

Although the concept of the arbitrariness of the sign was influential on modern linguistics, there are many evident cases where it does not hold which have attracted great interest among modern researchers. Cases of *sound symbolism* in language, where the sounds themselves in a word have some connection to what they describe, include onomatopoeic phrases such as *kapow* (a punch or banging sound) and *glub-glub* (water bubbling). Beyond direct imitation, it has been noted that English and other languages show correlations between certain phonetic structures and types of meaning, such as the array of English words beginning with *cr-* denoting brittleness (*crush*, *crunch*, *crash*, *crack*, *crackle*, *crinkle*, ...). This raises natural questions including how universal these patterns of iconicity are between languages and cultures, whether they are rooted in psychological tendencies that influence language or vice versa, what acoustic properties of speech sounds influence iconicity, and the extent to which less obvious iconic patterns shape language in general.

---

[1]via our project page `https://kiki-bouba.github.io/`

[2]"the sign is arbitrary"

37th Conference on Neural Information Processing Systems (NeurIPS 2023).

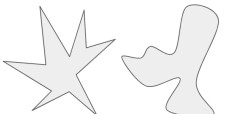 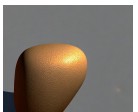 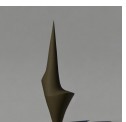 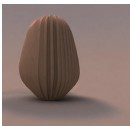 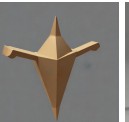 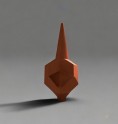 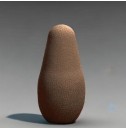

Original Experiment                   Random Image Generations

Figure 1: **Illustration of the kiki–bouba effect.** The shapes on the far left illustrate stimuli used in the classic *kiki–bouba* experiment. The remaining images are random generations from Stable Diffusion with the prompt *a 3D rendering of a ⟨w⟩ shaped object*, where ⟨w⟩ ∈ {*kiki*, *bouba*}. Which of of these images do you think were generated using pseudoword *kiki* and which with *bouba*? See below[3] for the answer.

Perhaps the most well-known and conclusively demonstrated examples of sound symbolism is the *kiki–bouba effect*. In this experiment, subjects are shown a spiky object and a round object, such as the left-hand shapes in Figure 1. When asked to assign one the name *kiki* and the other the name *bouba*, subjects show an overwhelming preference (see footnote[4] for the preferred matching). This effect has been shown for both auditory stimuli [19] and for written text (which implicitly maps to speech sounds) [13, 10, 14]. Furthermore, an array of studies have demonstrated it to hold across various speech sounds and constructed pseudowords (beyond *kiki* and *bouba*) [34], between different languages and cultures [3], and even among prelingual infants [40]. These findings challenge the Saussurean assumption that sound maps purely arbitrarily to meaning.

In another domain, recent years have seen explosive progress in the field of machine learning applied to natural language and vision, mainly powered by transformer neural networks and training on web-scale datasets of captioned images [6]. This includes discriminative models such as CLIP [45] and generative text-to-image models such as Stable Diffusion [49]. Although these models have revolutionized many applied tasks, they are largely used as a black box. One line of research examines the emergent behavior and inner structure of neural networks in order to interpret their behavior [48], and several works investigate the visual knowledge encoded by vision-and-language models (VLMs) in the context of human cognition and visual semantics in language [2, 61]. Given these parallels along with fundamental differences between these models and the human brain, it is natural to ask whether they have also learned to associate sounds in language (as reflected in their written representations) with visual semantics. In other words, does the vision-and-language setting reflect the presence of sound symbolic patterns in language? If so, this would provide strong evidence against Saussure's legendary claim from a novel computational perspective.

As seen in Figure 1, generative text-to-image models can be applied to text containing pseudowords such as *kiki* or *bouba*. Consider the various images generated by Stable Diffusion shown on the right-hand side of the figure. The round objects were generated with one of these pseudowords and the sharp objects were generated with the other (see below for the correct mapping), suggesting that the model associates each pseudoword with visual semantic properties such as roundness and sharpness. It is not obvious a priori that VLMs would learn associations with text input that is seemingly not well-formed; this suggestive parallel to human cognition invites methodical study. In this work, we analyze this phenomenon rigorously with quantitative as well as qualitative criteria, including generalizing to a larger set of constructed pseudowords reflecting many different speech sounds in English, and examining correlations between form and meaning on the grapheme (letter) as well as whole-word level.

To test for sound symbolism in VLMs, we use zero-shot knowledge probing which allows for evaluating the models' inherent knowledge—*i.e.,* not new knowledge acquired during further training. We leverage the ability of CLIP to embed text and image data in a shared semantic space in order to probe discriminative and generative (text-to-image) models with the same evaluation metrics. Our tests evaluate whether these models encode pseudowords similarly to humans with respect to known symbolic associations, comparing them to adjectives indicating properties related to "sharpness" and "roundness". To further ground our results in human cognition, we also conduct a user study testing the ability of subjects to reconstruct pseudowords used to condition text-to-image generations. Our results demonstrate that sound symbolism can indeed be observed in VLMs; the models under

---

[3]From left to right: *bouba, kiki, bouba, kiki, kiki, bouba*
[4]The vast majority of subjects prefer the name *kiki* for the spiky object and *bouba* for the round object.

consideration associate visual meanings with the written forms of particular speech sounds. Although we do not claim to answer exactly how this is learned by these models, our results suggest that VLMs could provide valuable insights regarding sound symbolism. In addition to providing a new perspective on this phenomenon in cognitive science, our work sheds light on what is learned by multimodal models in the context of interpretability in machine learning, demonstrating that these models also encode knowledge over individual characters beyond the semantics of words.

## 2    Related Work

**Sound symbolism in language.** A wide array of studies in cognitive psychology and linguistics have demonstrated the existence of sound symbolic associations, including the famous *kiki–bouba* experiment as well as many variations [51, 34, 28, 1, 7, 18, 10, 13, 17, 19]. This effect has been shown to be robust cross-culturally and across various languages and writing systems [4, 3, 14, 8], and even present among infants and toddlers [33, 40]. Corpus analyses of English have suggested that sound symbolic trends are present in the basic lexicon of the language – words with particular meanings may be statistically more likely to contain certain sounds [38, 60, 56]. Interestingly, research on blind individuals has found these associations to be weaker or absent when examined between spoken and tactile modalities, suggesting that visual input plays a key role [20, 22].

**Vision-and-language models.** Recent years have seen rapid development of foundation models, powerful pretrained models that can be adapted to downstream tasks, largely powered by transformer architectures and pretraining on web-scale data [6]. In the field of multimodal learning, significant foundation models include dual encoder models such as CLIP [45] and its open-source implementation OpenCLIP [11], trained with a contrastive objective on captioned image data. The power of these discriminative models for paired text-image understanding has been further leveraged in recent meteoric development of text-to-image generative models. Large diffusion models such as DALL-E 2 [47] and Imagen [50] set a new standard for general text conditioned image generation. Latent diffusion models such as Stable Diffusion make comparable results attainable with a smaller computational footprint by applying denoising diffusion to the latent space of an image autoencoder [49]. In particular, the classifier-free guidance [23] of Stable Diffusion uses OpenCLIP. In our work, we use OpenCLIP and Stable Diffusion as reference SOTA models for discriminative and generative text-image understanding respectively.

**Parallels between VLMs and cognitive science.** A number of prior works have examined the types of knowledge learned by artificial neural networks, with connections to learning and knowledge in cognitive science sometimes left implicit and sometimes stated explicitly. In particular, models trained on vision-and-language tasks have been found to encode knowledge about the visual world in ways that have parallels to human cognition. Although unimodal (text-only) language models learn some degree of factual information [44, 48, 12] and commonsense reasoning [59], the inclusion of visual data in training may bolster models with visual commonsense knowledge which is typically not written explicitly in text [58, 29, 26, 25, 61, 2]. Alper *et al.* [2] explicitly connect this to similar findings in cognitive science, where studies have explored the effect of human vision impairment and blindness on color associations [41, 53, 52, 55, 31]. In addition, Orgad *et al.* [39] note that text-to-image models in particular display implicit assumptions about the world based on correlations and biases in captioned image datasets.

Our work also explores the knowledge learned by VLMs, but rather than investigating the semantics of words or utterances in text, we focus on the *surface form* of textual input to these models and show that they have learnt a non-trivial mapping between sounds encoded by the written text and visual semantics. We also note that the surface form of textual input to these models has been explored in the context of typographic attacks [21, 32], in which models learn to associate text with images containing the text written out visually; however, they do not address semantic associations with the surface form of the input beyond its literal appearance as text.

## 3    Computational Paradigm for Sound Symbolic Probing

To test for the presence of sound symbolism in VLMs, we design a paradigm using controlled comparisons between pseudowords – nonsense words constructed from English letters with desired properties – and visual semantic properties. In particular, we are interested in whether a VLM

associates pseudowords with known sound-symbolic associations among humans (e.g. *kiki* – sharp) with visual properties such as "sharpness". For generative (text-to-image) models this translates to testing whether pseudowords containing letters that are known to have "sharp" associations tend to generate sharper images on average, and conversely for pseudowords containing letters with "round" associations. Recent progress in the field of multimodal learning makes it possible to test this directly, as CLIP [45] can be used to test images for semantic properties by comparing them to text prompts (e.g. *a 3D rendering of a sharp object*) via cosine similarity of embedding vectors in a semantic space shared between the textual and visual modalities. Moreover, CLIP can be tested directly as a discriminative model for sound symbolism by comparing the text embeddings of pseudowords to those of such properties.

This intuition is formalized below. We first describe how pseudowords with ground truth associations are constructed (Section 3.1). We then elaborate on our zero-shot knowledge probing techniques (Section 3.2) and evaluation protocol (Section 3.3).

### 3.1 Pseudoword Construction

Although studies using particular pseudoword stimuli such as *kiki–bouba* [46, p.19] or *maluma–takete* [24, p.133] have found that certain speech sounds show cross-modal associations, the precise nature of which acoustic phenomena give rise to these associations is an active topic of research. McCormick *et al.* [34] demonstrate that speech sounds form a spectrum from most "sharp" to most "round". Relevant phonetic distinctions include sonority and voicing in consonants, and vowel height and rounding. Fort and Schwartz [19] suggest that these properties can be aligned on the dimensions of "spectral balance and temporal continuity", which have also been found relevant for sharpness associations with non-speech sounds such as drum beats [19] and electronic sound waves [42].

To test for the presence of sound symbolism in the VLMs under consideration, we split speech sounds into "highly sharp" and "highly round" categories based on phonetic properties, leaving an investigation of more fine-grained distinctions within the spectrum of sounds to future research. Following the classification of speech sounds described in [34], we define the following subsets of English graphemes (letters) for use in our experiments, divided between consonants and vowels and split into classes based on their corresponding sounds:

$C_{☆}$: ⟨p t k s h x⟩    $C_{○}$: ⟨b d g m n l⟩
$V_{☆}$: ⟨e i⟩    $V_{○}$: ⟨o u⟩    $V_{-}$: ⟨a⟩

The classes above use ☆ and ○ as shorthand for "sharp" and "round" associations respectively, and $V_{-}$ indicates a neutral association (may appear in either class of pseudoword, as described below). The main phonetic distinctions used to determine these classes are voicing for consonants and backness for vowels; see the supplementary material for more information about the phonetic details motivating these classes.

We construct pseudowords for use in our experiments using the three syllable template $(CV)_1(CV)_2(CV)_1$ , where the first and last syllables are the same (e.g. *kitaki*, *bodubo*, . . . ). In addition, all graphemes must be drawn either from $C_{☆} \cup V_{☆} \cup V_{-}$ or $C_{○} \cup V_{○} \cup V_{-}$, disallowing forms like *kiduki* which mix graphemes from the two classes. We denote the set of all pseudowords with graphemes from the former set as $\Psi_{☆}$, the set of pseudowords with graphemes from the latter set as $\Psi_{○}$, and $\Psi = \Psi_{☆} \cup \Psi_{○}$. The choice of this template has a few motivations, including the large number of possible forms, few clashes with existing English words, and similarity to the *maluma–takete* stimuli used by Köhler [24, p.133]. Examples of such pseudowords include the following (in arbitrary order):

$\Psi_{☆}$:  *kitaki*   *hatiha*   *pepape*   *xisixi*   *hipehi*   *xaxaxa*   *texete*   ...   (324 total)
$\Psi_{○}$:  *gugagu*   *bodubo*   *gunogu*   *daluda*   *momomo*   *lunulu*   *gadaga*   ...   (324 total)

We also report results for the pseudowords *kiki* and *bouba* used verbatim, to reproduce the most well-known form of the *kiki–bouba* effect with VLMs. In accordance with the principles used to construct our pseudowords, *kiki* corresponds to class ☆ and *bouba* corresponds to class ○.

## 3.2 Zero-shot Knowledge Probing

We probe VLMs for sound symbolism with a zero-shot linear probe applied to embedding vectors in the multimodal embedding space of CLIP ($\subset \mathbb{R}^{1024}$). We consider only the zero-shot regime since we are interested in the inherent knowledge of our models (acquired from pretraining) and not the dynamics of training on new data. Furthermore, our approach tests VLMs end-to-end and is agnostic to the source of these effects with respect to the relative contribution of different model-internal components (such as tokenization and hidden activations).

**Prompts used**: We use the following prompts to probe the models under consideration, where $\langle w \rangle$ is the item (word or pseudoword) to be inserted into the prompt:

$P_1$: *"a 3D rendering of a $\langle w \rangle$ object"*
$P_2$: *"a 3D rendering of a $\langle w \rangle$ shaped object"*

We use $P_1$ for adjectives (e.g. *...of a round object*) and $P_2$ for nouns and pseudowords (e.g. *...of a cactus shaped object*, *...of a kiki shaped object*). These prompts are chosen for visual simplicity and interpretability; in the supplementary material we compare results on other prompts.

**Embeddings**: All of our tests use embedding vectors in CLIP space ($\subset \mathbb{R}^{1024}$) corresponding to pseudowords and adjectives inserted into prompts. These may either be text embeddings calculated using CLIP's text encoder, or image embeddings using CLIP's vision encoder applied to image generations conditioned on the given text. In either case, we embed pseudowords and adjectives in CLIP space to obtain vectors $v_{\langle w \rangle}, w_{\langle a \rangle} \in \mathbb{R}^{1024}$ respectively and unit-normalize them to $\hat{v}_{\langle w \rangle}, \hat{w}_{\langle a \rangle}$. When evaluating CLIP we directly encode both as text; when evaluating Stable Diffusion we calculate $v_{\langle w \rangle}$ using image generation, as detailed further in Section 4.1.

**Geometric scores** $\gamma_{\langle w \rangle}$: We propose a scoring method to measure geometric attributes such as sharpness or roundness, applied to images of objects or embeddings corresponding to pseudowords. To calculate this, we identify the one-dimensional semantic direction of interest in CLIP embedding space aligned with a collection of adjectives, similar to prior work on finding interpretable linear subspaces of word embeddings [35, 5, 16] and multimodal embedding spaces [57, 43]. We manually select 20 ground-truth adjectives, split evenly between those with sharp or round associations which we denote by $A_{\star}$ and $A_{\bigcirc}$ respectively. These are as follows:

$A_{\star}$ = {*sharp, spiky, angular, jagged, hard, edgy, pointed, prickly, rugged, uneven*}
$A_{\bigcirc}$ = {*round, circular, soft, fat, chubby, curved, smooth, plush, plump, rotund*}

Using the embeddings of these adjectives, we construct probe vector $w_{adj} := \sum_{\langle a \rangle \in A_{\star}} \hat{w}_{\langle a \rangle} - \sum_{\langle a \rangle \in A_{\bigcirc}} \hat{w}_{\langle a \rangle}$, unit-normalized to $\hat{w}_{adj}$. This vector approximates the direction in CLIP space representing the visual semantic dimension of interest; in other words, this is a unit vector pointing in the direction in CLIP's latent space which distinguishes between the two sets of adjectives. We probe item $\langle w \rangle$ by calculating the score $\gamma_{\langle w \rangle} := \hat{v}_{\langle w \rangle} \cdot \hat{w}_{adj}$, corresponding to projection onto the 1D subspace spanned by $w_{adj}$. Intuitively, the scalar $\gamma_{\langle w \rangle}$ measures whether the given item is closer to the "round" or "sharp" end of the scale of associations in our model's semantic space. Tangentially, while this work focuses on using these scores to place pseudowords along a sharp–round semantic axis, our geometric scoring method could be applicable for identifying words or images with such associations in more general settings as well.

**Phonetic scores** $\phi_{\langle w \rangle}$: We also propose a complementary scoring method to measure associations with categories of sounds reflected in English spelling. This is applied in order to quantify the phonetic or graphemic associations that our models show with real English words such as the adjectives given above. We construct probe vector $v_{pw} := \sum_{\langle w \rangle \in \Psi_{\star}} \hat{v}_{\langle w \rangle} - \sum_{\langle w \rangle \in \Psi_{\bigcirc}} \hat{v}_{\langle w \rangle}$, unit-normalized to $\hat{v}_{pw}$. This is a unit vector which estimates the dimension in CLIP space that best distinguishes between the two ground-truth classes of pseudowords. We then score item $\langle w \rangle$ via cosine similarity with this probe: $\phi_{\langle w \rangle} := \hat{v}_{pw} \cdot \hat{w}_{\langle w \rangle}$. Intuitively, the scalar $\phi_{\langle w \rangle}$ measures whether the given item (*e.g.*, adjective) is closer to pseudowords like *kitaki*$_{\star}$ or to those like *bodubo*$_{\bigcirc}$ in our model's semantic space. We call this "phonetic scoring" because it is a semantic dimension determined solely by the letters used in pseudowords and their underlying sounds.

### 3.3 Evaluation Method

For quantitative analysis, we evaluate the scores $\gamma_{\langle w \rangle}$ and $\phi_{\langle w \rangle}$ using classification metrics; $\gamma_{\langle w \rangle}$ for predicting the binary class ($\star$ or $\bigcirc$) of pseudowords, and $\phi_{\langle w \rangle}$ for predicting the binary class ($\star$ or $\bigcirc$) of adjectives with ground-truth labels. As these are unnormalized scores, we use non-probabilistic and threshold-agnostic metrics, namely ROC-AUC and Kendall correlation $\tau$ between the scores in question and the ground-truth classes as a binary indicator variable. These also have the desirable property of being symmetric with respect to our class labels, neither of which is privileged with respect to the other. Additionally, we provide an analysis of the correlation between $\gamma_{\langle w \rangle}$ and the particular sounds present in the first syllable of the given pseudoword.

We also reproduce the classic *kiki–bouba* experiment using the scores $\gamma_{kiki}$ and $\gamma_{bouba}$. To place these scores on an interpretable scale, we use their percentile ranks relative to scores of all pseudowords in $\Psi$, and report the difference in these percentiles $\Delta P_{kb}$. A high value for this metric indicates that the given model strongly aligns these pseudowords along the semantic axis of roundness and sharpness.

## 4 Results and Evaluation

In this section, we present our main findings. We first discuss details of the experimental setup (Section 4.1). We then present quantitative results (Section 4.2), the results of our user study (Section 4.3), and qualitative results (Section 4.4).

Further experimental details and results are provided in the supplementary material, including results on additional prompts and model architectures (such as a GAN-based text-to-image model). All of these settings show results consistent with our primary findings on CLIP and Stable Diffusion. There we also provide results for unimodal (text-only) text encoder models, which show mixed results when probed for sound symbolism.

Our primary findings in this work use models trained and prompted with texts in the English language. Nonetheless, in the supplementary material we also probe for sound symbolism in a multilingual text-to-image model with prompts in four geographically and linguistically diverse languages, with positive results. While we do not directly address the universality of sound symbolism across languages, these results warrant further investigation of sound symbolism in multilingual VLMs.

### 4.1 Experimental Details

We investigate the presence of sound symbolic patterns in two types of VLM: generative text-to-image models as exemplified by Stable Diffusion [49], and discriminative models with dual text and vision encoders as exemplified by CLIP [45]. We use the open-source CLIP implementation OpenCLIP [11] throughout; Stable Diffusion uses OpenCLIP as its text encoder for classifier-free guidance, motivating this comparison. We use these models as-is and probe them in the zero-shot regime, without any further training or adjustment of their tokenizer (which is identical for both models).

We note the a priori possibility that Stable Diffusion's weights may encode additional visual common-sense that is not present in CLIP. The majority of its parameters are in its UNet component (866M vs. 340M in its CLIP text encoder). These weights are trained on an image denoising task which requires understanding local regions in images, while CLIP's pretraining knowledge is only acquired from global image-text matching.

For CLIP, we assign embeddings to pseudowords by inserting them into textual prompts and embedding them with CLIP's text encoder. For Stable Diffusion, we embed pseudowords by generating images using such prompts and embedding the generated images using CLIP's vision encoder. Since the image generation process is stochastic, we reduce variance by generating multiple images with the corresponding prompt and calculating their mean embedding vector. As CLIP embeds text and images in the same space, we evaluate the pseudoword embeddings yielded by both models with the same probing method, as described in Section 3.3.

| Model | $\gamma_{\langle w \rangle}$ | | | $\phi_{\langle w \rangle}$ | |
|---|---|---|---|---|---|
| | AUC | $\tau$ | $\Delta P_{kb}$ | AUC | $\tau$ |
| Stable Diffusion | 0.74 | 0.34 | 80% | 0.97 | 0.68 |
| CLIP | 0.77 | 0.39 | 52% | 0.98 | 0.70 |
| (random) | 0.50 | 0.00 | 0% | 0.50 | 0.00 |

Table 1: **Results of zero-shot linear probing.** Results under $\gamma_{\langle w \rangle}$ indicate metrics for predicting pseudoword class (☆ or ◯) from geometric scores; results under $\phi_{\langle w \rangle}$ indicate metrics for predicting adjective class (☆ or ◯) from phonetic scores. Probing methods and evaluation metrics are as described in Section 3. (random) indicates the expected performance of a purely random scoring method, providing a lower bound for the performance of the models under consideration.

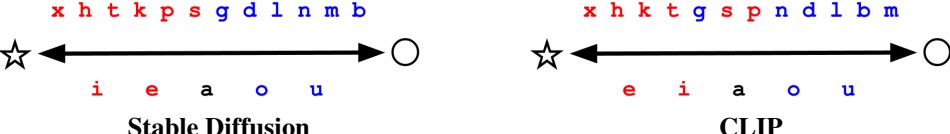

Figure 2: **Graphemes sorted by average geometric score** $\gamma_{\langle w \rangle}$ for pseudowords $\langle w \rangle$ whose first syllable contains the given grapheme, calculated with Stable Diffusion and CLIP. Characters are colored based on their ground-truth association (red for ☆, blue for ◯). Consonants are shown above and vowels below the arrow. We see that the two classes are mostly well-discriminated by these scores, especially when calculated Stable Diffusion. In this visualization, consonants and vowels are displayed on separate scales and are not positioned absolutely with respect to each other.

## 4.2 Quantitative Evaluation

We present the results of our quantitative tests in Table 1. Across all metrics, our probing methods are able to predict pseudoword and adjective classes (☆ or ◯) significantly better than chance. In particular, the results on Stable Diffusion indicate that images generated from pseudowords in $\Psi_{☆}$ are more likely to be visually "sharp" and images generated from pseudowords in $\Psi_{◯}$ are more likely to be visually "round", consistent with the examples shown in Figure 4. Additionally, we see that the $\Delta P_{kb}$ scores indicate a replication of the *kiki–bouba* effect in multimodal models, with *kiki* having a relatively higher affinity for "sharp" adjectives and the latter for "round" adjectives in such models.

In Figure 2, we see graphemes (letters), split into consonants and vowels and sorted by the average geometric score $\gamma_{\langle w \rangle}$ of all pseudowords containing them in the first syllable (first two characters). Surprisingly, we see an emergent pattern that closely reflects psychological and phonetic phenomena, as both models perfectly or nearly perfectly differentiate between ☆ and ◯-associated graphemes (as described in Section 3.1). This is despite the fact that these models were trained on text-image pairs on the full caption level, primarily saw valid English text during training, and did not have direct access to the auditory modality at all. We even see intriguing patterns within the intra-class

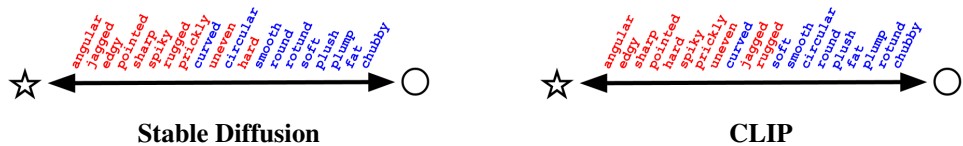

Figure 3: **Ground-truth adjectives sorted by phonetic score** $\phi_{\langle w \rangle}$, calculated with Stable Diffusion and CLIP. Adjectives are colored based on their ground-truth association (red for ☆, blue for ◯). We see that the two classes are highly differentiated by phonetic score for both models, as further reflected in the corresponding metrics in Table 1.

| POS | Lowest $\phi_{\langle w \rangle}$ ($\Rightarrow$ ◯) | Highest $\phi_{\langle w \rangle}$ ($\Rightarrow$ ☆) |
|---|---|---|
| **Stable Diffusion** | | |
| Noun | *butterball, yolk, pregnancy, booger, eggnog, turnip, bellyful, crybaby, doughboy* | *shard, kite, origami, hexagon, diamond, flake, octagon, triangle, protractor, lozenge, foldout* |
| Adj. | *obese, chubby, stinky, pudgy, overweight, fat, pregnant, plump, drowsy, soggy, squishy* | *triangular, diagonal, angular, shattering, jagged, rectangular, edgy, housebroken, geometrical* |
| **CLIP** | | |
| Noun | *doughboy, loudmouth, gumdrop, boogeyman, madwoman, lord, butterball, goddaughter* | *prefix, talkativeness, asker, flexibility, shears, shift, peek, slope, task, exit, hemline, tightness* |
| Adj. | *muggy, soggy, gloomy, grouchy, lumpy humongous, hungry, cloudless, unsmiling* | *apelike, flexible, diagonal, static, triangular, external, shipshape, interlocking, angular* |

Table 2: **Real English words sorted by phonetic score** $\phi_{\langle w \rangle}$. Results are split by part of speech (POS), from the ~$5.5K$ nouns in N and ~$1K$ adjectives in A. Columns indicate items with the lowest and highest scores out of the entire sorted lists, corresponding to more relative similarity to the pseudowords in $\Psi_{\bigcirc}$ or $\Psi_{\star}$ respectively. As seen above, these clearly depict "round" and "sharp" characteristics, particularly the words selected with the generative pipeline (*i.e.* Stable Diffusion).

ordering of graphemes, such as close vowels ⟨i u⟩ having more extreme association than close-mid vowels ⟨e o⟩ and voiced sonorants ⟨m n l⟩ having among the most "round" associations, although we leave investigation of such fine-grained patterns to future research.

Figure 3 shows the ground-truth adjectives from A$_\star$ and A$_\bigcirc$ sorted by phonetic score for both models, corresponding to the $\phi_{\langle w \rangle}$ metrics in Table 1. The near-perfect discrimination between the two classes under both models is consistent with the high metric values seen in the table.

## 4.3 User Study

In order to ground our results in human cognition, we conduct a user study using images produced by Stable Diffusion prompted using pseudowords. We adopt the two-alternative forced choice paradigm, where participants are provided with image pairs and corresponding pairs of pseudowords, and are asked to match the pseudowords to the corresponding images. We perform this study in two settings: in the first setting, we test participants' ability to distinguish between images generated using the specific pseudowords *kiki* and *bouba*; in the second setting, we test their ability to distinguish between images generated using random pseudowords from $\Psi_{\star}$ and $\Psi_{\bigcirc}$. One hundred subjects participated in the first setting, and seventy five participated in the second setting. Participants provided correct answers with overall accuracy of 73% in the *kiki–bouba* setting and 55% in the general pseudoword setting. In order to determine the statistical significance of these results while accounting for variation between subjects and items within each survey, we adopt a mixed-effects logistic regression model. We regress whether a question is answered correctly while treating question identity as a fixed effect and respondent identity as a random effect. Analysis of the first setting (*kiki* vs. *bouba*) results in an intercept estimate corresponding to a 89% overall success probability ($p < 0.001$); in the second setting (random pseudowords) this results in an intercept estimate corresponding to a 78% overall success probability ($p < 0.001$). By isolating the overall success rate from the effects of individual respondents and questions, our mixed-effects model results indicate that the sound symbolism exhibited by our text-to-image model correlates with human sound symbolic associations (with a stronger effect size in the simpler setting, but having a significant effect in both settings). Please refer to the supplementary material for additional details, including participant sourcing and compensation, the full text of instructions and survey examples, and a full statistical analysis of survey results.

## 4.4 Qualitative Results

We provide a qualitative analysis using real words from English, filtered to select for basic words and split by part of speech: ~$5.5K$ nouns denoted by N, and ~$1K$ adjectives denoted by A. We select basic concrete words for both categories by using lemmas from WordNet [36, 37], filtering out

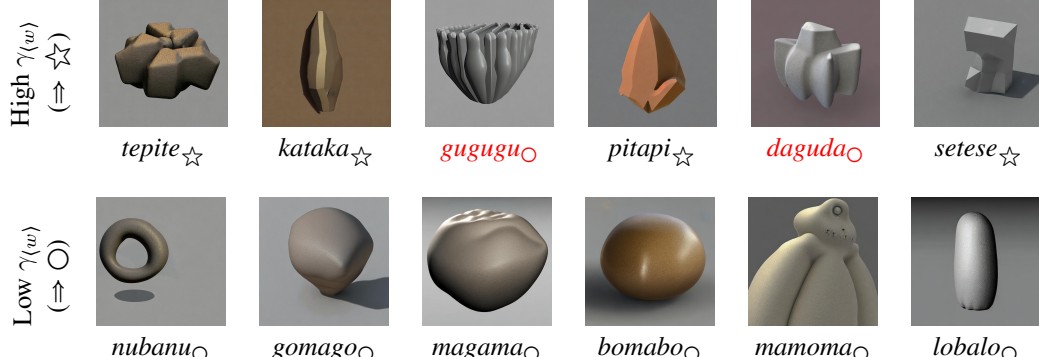

**Figure 4: Image generations for pseudowords with high (top 20%) and low (bottom 20%) geometric scores**. We visualize random selections of pseudoword–image pairs for each category. Pseudowords with class (☆ or ○) that does not match its geometric score are indicated in red. As seen above, the shapes of the generated images noticeably correlate with the pseudoword class.

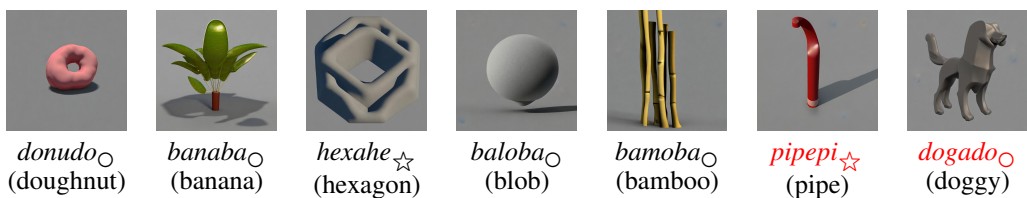

**Figure 5: Images generated from pseudowords reminiscent of real English words.** For each pseudoword we display an associated image generation and the automatically detected closest English word. Pseudowords with high or low geometric scores (in top or bottom 20% relative to all pseudowords) which do not match their class (☆ or ○) are indicated in red.

obscure or highly abstract items using age of acquisition and concreteness scores from [27, 9]. We sort these items by their $\phi_{\langle w \rangle}$ scores and display the words with the highest and lowest scores.

Qualitative results are shown in Table 2 and Figure 4. In Table 2, we see a striking pattern that the head and tail of this list represent adjectives which intuitively describe visual properties that are more "sharp" or "round". In other words, these properties are highly correlated with relative similarity to the two pseudoword classes ○ and ☆. This strengthens our quantitative results for ranking ground-truth adjectives with phonetic scores $\phi_{\langle w \rangle}$, as we see these scores are truly aligned with visual semantic properties for the VLMs under consideration (particularly for Stable Diffusion). The images in Figure 4 show a similarly striking pattern, with the sharper objects (as measured by $\gamma_{\langle w \rangle}$ score) being more commonly generated by pseudowords from $\Psi_{\bigstar}$ and rounder objects by pseudowords from $\Psi_{\bigcirc}$.

Pseudowords that resemble real English words may generate images reminiscent of real objects, as seen in Figure 5. There we display such pseudowords along with close English words, detected automatically via an automatic search heuristic (described in the supplementary material) combining fuzzy string matching and CLIP text-image similarity. This suggests a possible correlation between sounds in real English words and visual semantics as explored in the psycholinguistic literature [38, 60, 56].

## 5 Discussion, Limitations and Future Work

We have shown that sound symbolism is reflected in VLMs by evaluating them on pseudowords with controlled phonetic properties. By comparing pseudowords built from known "sharp" or "round" speech sounds, we find that these models learn associations with corresponding sharp and round adjectival properties, parallel to the classic *kiki–bouba* experiment in psychology.

Further strengthening these findings, we provide strong evidence in the supplementary material that these models have not learned specifically from items illustrating the *kiki–bouba* effect, by showing that this concept is not well-represented in the LAION dataset [54] upon which our VLMs were

trained, and by showing that Stable Diffusion does not generate coherent content from prompts referring to the experiment itself. Rather, it appears that our VLMs have learned sound patterns from their training data via large-scale correlations between sounds and shapes. Future work could further explore complex generalization patterns and biases learned by VLMs, associating non-obvious visual or semantic meanings with particular input patterns.

Our findings have significance both in the field of multimodal machine learning as well as in psycholinguistics. From a computational perspective, our work reveals an emergent behavior in VLMs that has been hitherto unexplored, to the best of our knowledge. In general, VLMs are being used as black boxes without a full understanding of how they understand the visual semantics of language. Our work sheds light on how these models interpret and respond to language, as well as raising questions about how sound symbolic associations are inferred from our models' training datasets, whether additional dimensions of sound symbolic meaning are present in these models apart from our findings, and whether similar associations could exists with other modalities beyond vision.

From a cognitive science perspective, our work provides a new perspective to the extensive literature on sound symbolism, which has long been a topic of interest and debate. If VLMs learn sound symbolic association from valid text in caption data, this may be due to the presence of sound symbolism as reflected in the basic lexicon of English (or other languages), and the presence of sound symbolism in language itself is moderately controversial (notably denied by Ferdinand de Saussure as discussed in Section 1). In this vein, investigation into how VLMs infer sound symbolic associations from their training data could potentially shed light on how sound symbolism is learned during human language acquisition. Additionally, these methods could provide new insights into the classic questions of what aspects of sound are tied to meaning and to what extent the observed phenomena are culturally universal or specific to the English language. Our work provides a new line of inquiry, suggesting that VLMs and their training datasets should be researched carefully for further insight.

Regarding limitations of our findings, we do not assume that these models directly imitate human cognition, although our results do show that sound symbolic patterns are present in captioned image data with a strong parallel to the psychology of language and vision. Additionally, we do not answer *why* these associations are present in language or exactly how they are learned from the given data. In particular, we note that our results are agnostic to whether these models have memorized associations between particular letters and shapes, and to whether they have a deeper understanding of the phonetic or acoustic properties implied by these letters. We leave investigation of these questions to future work, and foresee them providing an important direction for research into sound symbolism in human cognition and machine learning.

## Acknowledgments and Disclosure of Funding

This work was partially supported by the Alon Fellowship. We thank Gal Fiebelman and Taelin Karidi for their helpful feedback.

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
