# Kiki or Bouba? Sound Symbolism in Vision-and-Language Models —Supplementary Material—

**Morris Alper and Hadar Averbuch-Elor**
Tel Aviv University

## Contents

## 1 Background on Phonetics and Sound Symbolism

In this section, we provide background on phonetics of the English language, with an emphasis on the distinctions relevant to our work. Following the convention in modern linguistics, we use the

37th Conference on Neural Information Processing Systems (NeurIPS 2023).

International Phonetic Alphabet (IPA) to represent *phonemes* (units of sound) and write IPA symbols between slashes /.../, while we write graphemic (written) representations in angle brackets ⟨...⟩. Unless stated otherwise we use American English for transcriptions. For more thorough overviews of phonetics, the IPA, and the sounds used in English and across the languages of the world, see [6, 1].

In general, speech sounds may be roughly split into the categories of *consonants* and *vowels*, based on the degree of closure of the vocal tract.

For consonants, McCormick *et al.* [10] find that the most round-associated segments are all *voiced* (sonorants /m n l/ and voiced stops /b d g/). In this context, *voicing* refers to vibration of the vocal cords during the production of a phoneme[1]. On the other end of the spectrum, they find a sharper association for the voiceless stop consonants /p t k/, and for the *fricative* and *affricate* consonants /f v s z tʃ dʒ/, the latter produced by making a partial occlusion in the vocal tract to produce turbulent airflow.

Among vowels, McCormick *et al.* find the dimension of *roundedness* (in an articulatory sense) to be significant for sound symbolic association among vowels, with rounded vowels being more round-associated. In the context of articulatory phonetics, *roundedness* refers to a vowel produced with lips pursed together. In English this is coupled with *backness* which roughly refers to the position of the tongue relative to the back of the mouth; for example /ɛ/ (as in *bell*) is a front unrounded vowel while /o/ (as in *bowl*) is a back rounded vowel.

Our split of English graphemes (letters) into ☆ and ◯-associated categories for the purpose of constructing pseudowords is motivated by this phonetic background, along with a desire for balanced class sizes and avoiding digraphs (e.g. ⟨th⟩, which represents a single phoneme in English). For consonants, our ☆-associated graphemes ⟨p t k s h x⟩ are all voiceless, fricatives or affricates while ◯-associated ⟨b d g m n l⟩ are all voiced. For vowels, although English vowel orthography has a complex mapping to sound segments, we treat ⟨e i⟩ as front unrounded vowels (and hence ☆) and ⟨o u⟩ as back rounded vowels (and hence ◯) as a simplifying assumption, consistent with prior work. Since the grapheme ⟨a⟩ can correspond to both front (/æ/ as in *fat*) and back (/ɑ/ as in *father*) vowels in English, we treat it as neutral with respect to sharpness associations[2].

## 2   Experimental Details

### 2.1   Image Generation Settings

To generate images with Stable Diffusion, we use guidance scale 9 and 20 inference steps with DPM-Solver++ [8, 9] as implemented in the Hugging Face `diffusers` library as `DPMSolverMultistepScheduler`. All images were generated using minibatches of 50 generated images for each prompt; 50 is chosen empirically based on the tradeoff between variance reduction (when averaging image embeddings) and computational constraints. All image generations use random seeds. In all figures shown in the main paper, the image displayed for each prompt is that closest to the centroid of the images in its respective minibatch (in OpenCLIP space, using embedding similarity).

### 2.2   Model Checkpoints, Settings, and Compute

We use the Hugging Face `transformers` [23] (v4.27.3) and `diffusers` [14] (v0.14.0) APIs for loading the models under consideration from the following checkpoints:

- OpenCLIP: `laion/CLIP-ViT-H-14-laion2B-s32B-b79K`
- Stable Diffusion: `stabilityai/stable-diffusion-2`

The only compute-intensive step in our method is producing image generations for every pseudoword. We use a single NVIDIA RTX A5000 GPU and are able to fit generation of a minibatch of 50 images in GPU memory by using attention slicing and VAE slicing, which reduce GPU memory consumption

---

[1]Strictly speaking, phonemically "voiced" obstruents such as /b/ may be optionally devoiced in English making the relevant distinction one of aspiration, but this level of phonetic detail is not relevant for our discussion.

[2]This is also consistent with previous work, which has used ⟨a⟩ in both "round" stimuli like *maluma* and "sharp" stimuli like *takete*.

| Model | $\gamma_{\langle w \rangle}$ $\tau$ | $\phi_{\langle w \rangle}$ $\tau$ |
|---|---|---|
| Stable Diffusion | 0.34 ($p < 10^{-25}$) | 0.68 ($p < 10^{-3}$) |
| CLIP | 0.39 ($p < 10^{-32}$) | 0.70 ($p < 10^{-3}$) |

Table 1: **Kendall correlation scores along with p-values** for models under consideration, demonstrating statistical significance. Interpretation of these p-values is discussed in Section 2.4.

at the expense of inference speed. With these settings, generation uses approximately 23GB of GPU memory and generating a single minibatch takes about 6 minutes.

## 2.3 Construction of Word Lists

The lists of real English words used for qualitative evaluation—~$5.5K$ nouns denoted by N, and ~$1K$ adjectives denoted by A—were constructed as follows. We first select all lemmas of synsets from WordNet [11, 12] with noun and adjective part of speech labels respectively, along with the remaining adjectives from $A_\star$ and $A_\bigcirc$. In order to avoid obscure and highly abstract words, we filter these lists using age of acquisition (AoA) and concreteness scores from the word lists of Kuperman *et al.* [5] and Brysbaert *et al.* [2]. In particular, we remove words with AoA $\geq 10$ and concreteness score $\leq 2.5$, along with words that do not appear in these datasets. We also manually remove a handful of obscenities and inappropriate items.

Examples of items from these lists include the following randomly-selected words:

N:  *collarbone, poncho, lip, baseball, germ, swordsman, bumpiness, hitter, pilgrim*
A:  *electrical, shy, antiseptic, hearty, snappy, moist, lifeless, skinny, solitary, floury*

## 2.4 Statistical Significance

In Table 1, we reproduce the Kendall correlation metrics from our main paper along with upper bounds on p-values. These are available since the Kendall correlation $\tau$ can be interpreted as a non-parametric test statistic, and we calculate these p-values using the `scipy.stats.kendalltau` function from the `scipy` library. We note that all of these results are statistically significant even at the very low significance threshold $\alpha = 10^{-3}$, indicating that these metric values would be highly unlikely under the null hypothesis in which there would be no underlying association between the quantities being compared.

## 3 Additional Results

### 3.1 Additional Multimodal Models

We provide results on additional text-conditioned image generation models, covering various model architectures, pretraining data and input languages. We consider the SOTA text-to-image diffusion models DeepFloyd-IF [19] and Kandinsky [15], as well as the SOTA text-conditioned GAN model GALIP [21]. To evaluate these models, we use the methodology from our main paper. This includes prompts and evaluation metrics, calculated by embedding generated images with CLIP. We use the following model checkpoints from Hugging Face Model Hub to evaluate DeepFloyd-IF and Kandinsky: `DeepFloyd/IF-I-M-v1.0`, `kandinsky-community/kandinsky-2-1-prior`, `kandinsky-community/kandinsky-2-1`. For GALIP, we use the official code implementation with the checkpoint trained on the CC12M dataset.

We use the following inference settings: We run DeepFloyd-IF and Kandinsky in half-precision. For DeepFloyd-IF, we only run the first stage of inference which outputs low-resolution ($64 \times 64$) images. For Kandinsky, we use both prior and image-to-text pipelines with guidance scale set to 1.0 and $768 \times 768$ output. All other inference settings use the defaults from the model checkpoint configuration files and the `DiffusionPipeline` class from the Hugging Face `diffusers` library. For GALIP, we use the default settings provided in the inference notebook in its official repository.

|  | $\gamma_{\langle w \rangle}$ | | | $\phi_{\langle w \rangle}$ | |
| --- | --- | --- | --- | --- | --- |
| Model | AUC | $\tau$ | $\Delta P_{kb}$ | AUC | $\tau$ |
| DeepFloyd-IF | 0.63 | 0.18 | 27% | 0.98 | 0.70 |
| Kandinsky | 0.59 | 0.12 | 36% | 0.85 | 0.51 |
| GALIP | 0.62 | 0.17 | 41% | 0.98 | 0.70 |
| Stable Diffusion | 0.74 | 0.34 | 80% | 0.97 | 0.68 |
| CLIP | 0.77 | 0.39 | 52% | 0.98 | 0.70 |
| (random) | 0.50 | 0.00 | 0% | 0.50 | 0.00 |

Table 2: **Quantitative results for additional multimodal models.** We report results for additional SOTA text-to-image models with various architectures and pretraining data sources. We also report the results for the vision-and-language models considered in our work for comparison, as well as the random baseline indicated by (random).

For all of these models, we generate a single image for each pseudoword; for "kiki" and "bouba" generations to calculate $\Delta P_{kb}$, we generate 10 images for each for each.

In Table 2, we show quantitative results for these models, which all show significant sound symbolic effects. We note that, in addition to architectural differences from Stable Diffusion, these models are all trained on different datasets – in particular, GALIP was not trained on LAION data at all, precluding the data distribution in LAION as the only source of the observed effects.

### 3.2 Unimodal Models

As an additional comparison, we use our same methodology to probe unimodal text encoder models for sound symbolic associations. Since our metrics rely on cosine similarity of pooled embeddings, we use Sentence Transformer models [16] which were trained with a cosine similarity-based contrastive semantic objective. In particular, we compare the following models:

- SMPNet: MPNet [20] fine-tuned with a Sentence Transformer objective (Hugging Face checkpoint `sentence-transformers/all-mpnet-base-v2`, 109M parameters)

- SDRoBERTa: DistilRoBERTa [18] fine-tuned with a Sentence Transformer objective (Hugging Face checkpoint `sentence-transformers/all-mpnet-base-v2`, 82M parameters)

- SMiniLM: MiniLM [22] fine-tuned with a Sentence Transformer objective (Hugging Face checkpoint `sentence-transformers/all-MiniLM-L12-v2`, 33M parameters)

- SALBERT: ALBERT [7] fine-tuned with a Sentence Transformer objective (Hugging Face checkpoint `sentence-transformers/paraphrase-albert-small-v2`, 2M parameters)

Quantitative results are shown in Table 3. Interestingly, the larger unimodal models SMPNet and SDRoBERTa show stronger sound symbolic associations, while the smaller SMiniLM and SALBERT models show near-random performance by most metrics. We also show qualitative results for SMPNet, the unimodal model with overall highest sound symbolic metrics, in Table 4. There it can be seen that the model appears to partially reflect "sharp" and "round" semantic associations using our phonetic scoring method, but also partially reflects the surface spelling of the English words being evaluated. These mixed results raise questions regarding the ability of unimodal models to learn sound symbolic associations from text data alone and the relative contributions of textual and image data to this phenomenon; we leave a thorough investigation of these topics to future work.

### 3.3 Additional Prompts

We provide a comparison of results using various prompts with CLIP in Table 5, with the prompt used in our main paper (*a 3D rendering of a $\langle w \rangle$ object*) in the final row. In all cases, we use the prompt as-is when inserting adjectives, and with the word "shaped" added (e.g. *a 3D rendering of a $\langle w \rangle$ shaped object*) when inserting nouns and pseudowords. As can be seen there, all prompts under consideration display sound symbolic effects despite some variation in metric values.

| | $\gamma_{\langle w \rangle}$ | | | $\phi_{\langle w \rangle}$ | |
|---|---|---|---|---|---|
| Model | AUC | $\tau$ | $\Delta P_{kb}$ | AUC | $\tau$ |
| SMPNet | 0.76 | 0.37 | 42% | 0.82 | 0.46 |
| SDRoBERTa | 0.74 | 0.34 | 62% | 0.71 | 0.30 |
| SMiniLM | 0.53 | 0.04 | 60% | 0.57 | 0.10 |
| SALBERT | 0.50 | 0.01 | 44% | 0.54 | 0.06 |
| Stable Diffusion | 0.74 | 0.34 | 80% | 0.97 | 0.68 |
| CLIP | 0.77 | 0.39 | 52% | 0.98 | 0.70 |
| (random) | 0.50 | 0.00 | 0% | 0.50 | 0.00 |

Table 3: **Quantitative results for unimodal models.** We calculate the metrics used in our main paper for four different unimodally (text-only) trained models, all fine-tuned with the Sentence Transformer objective which makes cosine similarity-based probing semantically meaningful for text encoders. We also report the results for the vision-and-language models considered in our work for comparison, as well as the random baseline indicated by (random).

| POS | Lowest $\phi_{\langle w \rangle}$ ($\Rightarrow$ ◯) | Highest $\phi_{\langle w \rangle}$ ($\Rightarrow$ ☆) |
|---|---|---|
| **SMPNet** | | |
| Noun | *baboon, mongoose, moo, hobo, bogeyman, gab, bedbug, loon, gremlin, ladybug, noodle* | *hexagon, tyke, skateboarder, pillar, pinstripe, hive, yoke, pinecone, teakettle, pane, whiskers, star* |
| Adj. | *grizzly, lunar, beaded, moonless, bluish, muddy, gooey, wormy, moonlit, blubbery* | *khaki, shipshape, teensy, twinkly, bladed, starlit, pointed, teenage, whiskered, scaly, spiky* |

Table 4: **Real English words sorted by phonetic score for SMPNet,** the unimodal model under consideration with highest overall sound symbolic metrics. As seen above, the order of words partially matches intuition about "round" and "sharp" characteristics, but also appears to moderately correlate with the initial letters in the English words (e.g. many words in the left column begin with ◯-associated letters *b-*. *m-*, *g-*, or *l-*).

## 3.4 Multilingual Results

To investigate whether sound symbolism may exist in a multilingual VLM, we evaluate the Kandinsky [15] multilingual text-to-image model with our methodology on prompts in multiple languages. While we evaluate Kandinsky on English prompts in Section 3.1, this model can also receive multilingual prompts (and uses multilingual CLIP as its text encoder). In particular, we construct prompts (shown in Table 6) in four geographically and linguistically diverse languages: Finnish, Indonesian, Hungarian, and Lithuanian. Using the same inference methodology as in Section 3.1, we evaluate Kandinsky on these prompts; results are shown in Table 7. We find non-trivial sound symbolism in this setting in each language, suggesting that sound symbolism may be learned in a multilingual vision-and-language setting.

We place these results in context by emphasizing that we do not claim to demonstrate the universality of sound symbolism across languages; rather, our work work focuses on showing the that common VLMs are observed to learn sound symbolic associations. Nevertheless, these positive results in a multilingual setting are suggestive and further investigation of multilingual vision-and-language models might provide insight into cross-linguistic sound symbolic patterns in language.

## 3.5 Corner Detection Probing

To provide an additional visually grounded probe for sound symbolism in text-to-image models, we estimate the sharpness or roundness of images with a corner detection algorithm (as more corners generally correspond to a visually more "sharp" image). In particular, we apply the Harris corner detector [3] to our pseudoword image generations produced by Stable Diffusion. We use the `cornerHarris` implementation in the `OpenCV` library with parameters $(5, 15, 0.04)$ to our images (of size $768 \times 768$) and set negative output values to zero. We then apply non-maximum suppression

| Prompt | $\gamma_{\langle w \rangle}$ | | | $\phi_{\langle w \rangle}$ | |
| --- | --- | --- | --- | --- | --- |
| | AUC | $\tau$ | $\Delta P_{kb}$ | AUC | $\tau$ |
| *a $\langle w \rangle$ object* | 0.78 | 0.40 | 32% | 0.96 | 0.67 |
| *a picture of a $\langle w \rangle$ object* | 0.83 | 0.46 | 67% | 0.99 | 0.71 |
| *an oil painting of a $\langle w \rangle$ object* | 0.75 | 0.35 | 32% | 1.00 | 0.73 |
| *a $\langle w \rangle$ thing* | 0.82 | 0.45 | 35% | 0.98 | 0.70 |
| *a $\langle w \rangle$ item* | 0.75 | 0.35 | -35% | 0.92 | 0.61 |
| *a $\langle w \rangle$ drawing* | 0.84 | 0.48 | 17% | 0.95 | 0.65 |
| *this thing is $\langle w \rangle$* | 0.79 | 0.41 | 59% | 0.91 | 0.59 |
| *$\langle w \rangle$* | 0.77 | 0.39 | 18% | 0.93 | 0.62 |
| *a 3D rendering of a $\langle w \rangle$ object* | 0.77 | 0.39 | 52% | 0.98 | 0.70 |

Table 5: **Results on various prompts**, evaluated using CLIP with our probing methods. The prompt used in our main paper is reported in the last row. Although the metric values vary somewhat by prompt, all of the prompts under consideration exhibit sound symbolic effects.

| Language | Prompt |
| --- | --- |
| Finnish | *3D-renderöinti objektista, jolla on muoto: "$\langle w \rangle$"* |
| Indonesian | *rendering 3D objek dengan bentuk: "$\langle w \rangle$"* |
| Hungarian | *egy objektum 3D-s megjelenítése alakzattal: "$\langle w \rangle$"* |
| Lithuanian | *3D objekto atvaizdavimas su forma: "$\langle w \rangle$"* |

Table 6: **Multilingual prompts** used for results in Table 7. $\langle w \rangle$ indicates the slot where pseudowords are inserted. Each prompt roughly translates to English *a 3D rendering of an object with shape: "$\langle w \rangle$"*, chosen for cross-linguistic grammatical uniformity.

using a $100 \times 100$ sliding window. Finally, we calculate the maximum output value $M$ and count the number of outputs that are greater than $0.01 * M$. This yields the number of corner detections in the given image.

Applying this to Stable Diffusion generations for pseudowords, we find that "round" and "sharp" pseudowords correspond to $11.47$ and $12.75$ corners per image on average, respectively. To determine the significance of this effect, we apply a two-sided Welch's t-test to the number of corners per image for the 32.4K image generations (50 per pseudoword), testing for a significant difference in the mean number of corners per image between the "sharp" and "round" association classes (each comprising exactly half of the items). This test yields t-statistic $12.028$ with $p < 10^{-32}$, indicating a significant difference between the means of the two classes. In other words, we find a significant difference in the number of corner detections for images generated from "sharp" and "round" pseudowords, confirming their visual distinctness on average.

### 3.6 Kiki–Bouba in LAION

We first manually examine image generations for various prompts that directly refer to the *kiki–bouba* effect, to evaluate whether Stable Diffusion (trained on the LAION dataset) shows signs of recognizing this as a concept. We generate minibatches (50 images each) of images for a variety of such prompts, listed below. From visual inspection, none of these prompts yield images with a coherent relation to the *kiki–bouba* effect. Examples of full minibatches of image generations for some of these prompts are shown in Figure 1.

Prompts manually examined include: *a picture of the kiki-bouba experiment, a picture of the kiki/bouba experiment, a picture of the bouba-kiki experiment, a picture of the bouba/kiki experiment, a picture of kiki and bouba, a picture of bouba and kiki, kiki and bouba, bouba and kiki, a picture of the kiki-bouba effect, a picture of the kiki/bouba effect, a picture of the bouba-kiki effect, a picture of the bouba/kiki effect, shapes for the bouba-kiki experiment, shapes for the bouba-kiki effect, shapes for the kiki-bouba experiment, shapes for the kiki-bouba effect, bouba and kiki psychological stimuli,*

| Language | $\gamma_{\langle w \rangle}$ | | | $\phi_{\langle w \rangle}$ | |
|---|---|---|---|---|---|
| | AUC | $\tau$ | $\Delta P_{kb}$ | AUC | $\tau$ |
| Finnish | 0.69 | 0.27 | -4% | 0.94 | 0.64 |
| Indonesian | 0.67 | 0.24 | 4% | 0.94 | 0.64 |
| Hungarian | 0.60 | 0.14 | 9% | 0.93 | 0.62 |
| Lithuanian | 0.73 | 0.32 | 23% | 0.97 | 0.68 |
| (random) | 0.50 | 0.00 | 0% | 0.50 | 0.00 |

Table 7: **Multilingual results** for generations, using the Kandinsky text-to-image model with prompts from Table 6. (random) indicates the expected performance of a purely random scoring method. Recall that the AUC and $\tau$ metrics are calculated over our full set of 648 pseudowords, and $\Delta P_{kb}$ is calculated using "kiki" and "bouba" alone.

*the shapes bouba and kiki as used in psychological research, the bouba/kiki stimuli in psychology, bouba and kiki shapes as known from linguistics*

These results suggest that this concept is not readily learnable from the LAION dataset; to further strengthen this hypothesis, we perform a search of the LAION dataset, using the ~$2B$ LAION-2B subset of primarily English captions available on the Hugging Face datasets hub as `laion/laion2B-en`[3]. We use caption data alone (without the corresponding images) and only consider items indicated as non-NSFW in the accompanying metadata; we preprocess by converting captions to lowercase. Searching for captions containing both *kiki* and *bouba* as substrings and excluding those containing the West African name *Boubacar*, we are left with only the following 12 captions in the entire dataset:

- *the bouba kiki effect and language are there certain human sounds with meanings that can cross the language in the bouba-kiki effect, and that you want to know if the picture is bouba or kiki.*
- *kiki or bouba: what is the shape of your taste?*
- *ramachandran and hubbard [3] suggest that the kiki/bouba effect has implications for the evolution of language, because it suggests that the naming of*
- *preschool-age sound- shape correspondences to the bouba-kiki effect karlee jones, b.s. ed. & matthew carter, ph.d. valdosta state university.*
- *the bouba-kiki effect*
- *the bouba kiki effect and language if you were looking at two shapes—specifically, a pointy, jagged polygon and an amoeboid-like splotch—which would you name bouba, and which would you name kiki.*
- *grooving: kiki and bouba minds*
- *bumblebee and how to design the transformers || the kiki/bouba effect*
- *the bouba/kiki effect recent work by daphne maurer and colleagues has shown that even children as young as 2.5 (too young to read) show this effect.*
- *the bouba / kiki effect*
- *the bouba kiki effect and language if you were looking at two shapes—specifically, a pointy, jagged polygon and an amoeboid-like splotch—which would you name bouba, and which would you name kiki.*
- *the bouba/kiki effect the bouba/kiki effect was first observed by german-american psychologist wolfgang köhler in 1929.*

The learnability of rare concepts for text-to-image generation has been explicitly studied by Samuel *et al.* [17], who find that Stable Diffusion struggles to accurately depict concepts with less than 10,000 samples in LAION-2B. Therefore we conclude that it is highly unlikely that Stable Diffusion has direct knowledge of the *kiki–bouba* effect as an abstract concept.

---

[3]`https://huggingface.co/datasets/laion/laion2B-en`

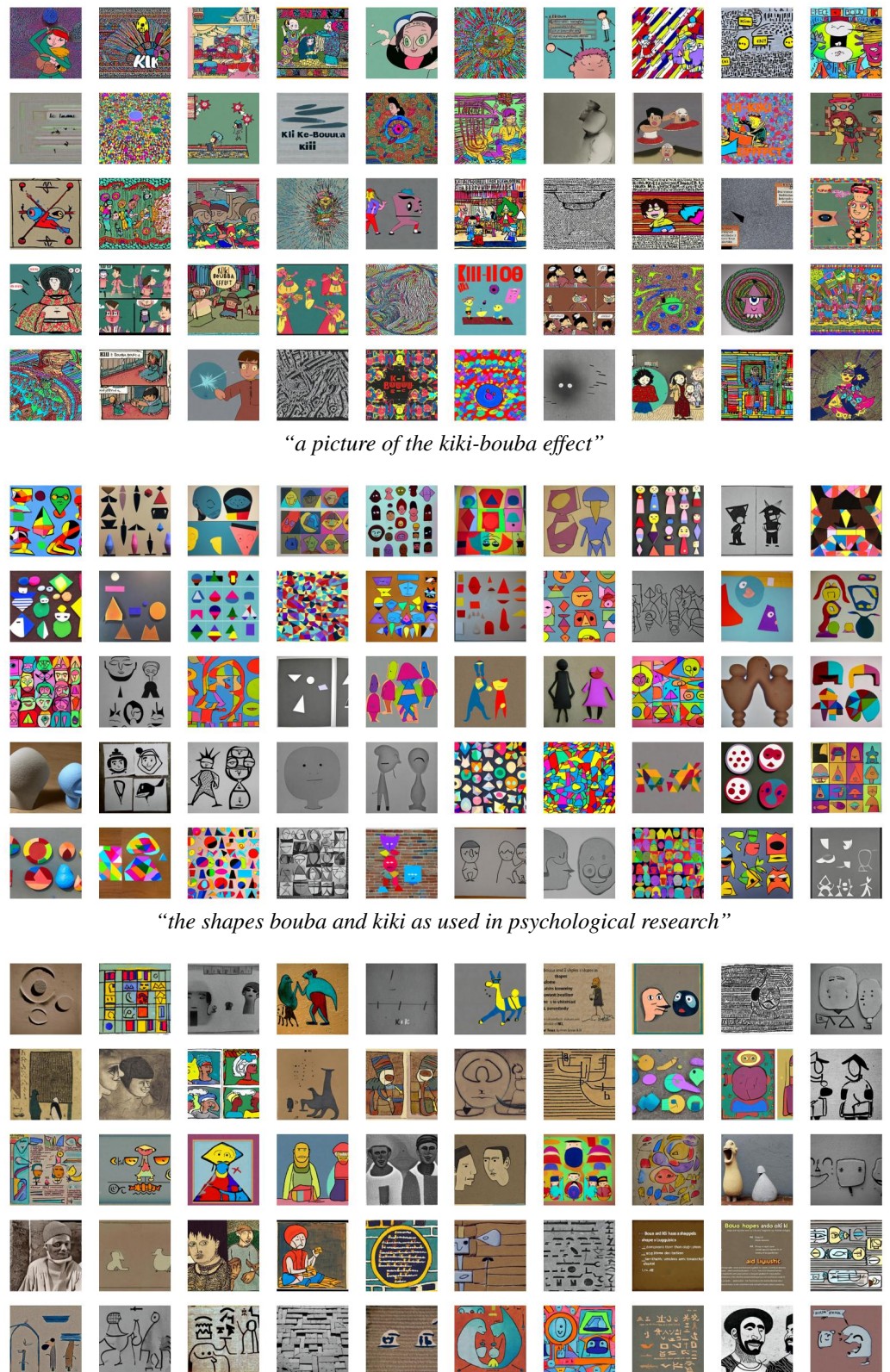

*"a picture of the kiki-bouba effect"*

*"the shapes bouba and kiki as used in psychological research"*

*"bouba and kiki shapes as known from linguistics"*

Figure 1: **Image generations for prompts describing the kiki–bouba effect.**. We display all images from a single (size 50) minibatch of text to image generations for each of the given prompts describing the *kiki–bouba* effect. The generations do not coherently depict the psychological effect as a concept.

### 3.7 Pseudowords Resembling Real English Words

In order to automatically find the English word of which a given pseudoword is reminiscent (e.g. *donudo* – "doughnut"), we use the following heuristic algorithm. Given English word $\langle w \rangle$ and pseudoword $\langle p \rangle$, we calculate:

1. $s_{clip}(\langle w \rangle, \langle p \rangle)$: The cosine similarity of CLIP embeddings of the word $\langle w \rangle$ (as text) and the mean embedding of images generated using the pseudoword $\langle p \rangle$.

2. $s_{text}(\langle w \rangle, \langle p \rangle)$: Text similarity score ($\in [0, 1]$) based on Levenshtein edit distance [13]. In particular, let $|\langle w \rangle|$ and $|\langle p \rangle|$ be the length in characters of $\langle w \rangle$ and $\langle p \rangle$ respectively, $d$ the edit distance between $\langle w \rangle$ and $\langle p \rangle$, and $m = \max\{|\langle w \rangle|, |\langle p \rangle|\}$; then we define $s_{text} := 1 - d/m$.

We define the score $s(\langle w \rangle, \langle p \rangle) := s_{clip}(\langle w \rangle, \langle p \rangle) \cdot s_{text}(\langle w \rangle, \langle p \rangle)$. Intuitively, this is maximized when $\langle w \rangle$ is both textually similar to $\langle p \rangle$ and semantically matches the images generated from $\langle p \rangle$. On input $\langle p \rangle$, our heuristic returns $\arg\max_{\langle w \rangle} s(\langle w \rangle, \langle p \rangle)$, the English word maximizing this score. We constrain this search to basic, concrete English words by filtering with age of acquisition (AoA) and concreteness scores found in the word lists of Kuperman *et al.* [5] and Brysbaert *et al.* [2]; in particular, we use the $\sim 8.3K$ English words from these lists with AoA < 10 and concreteness < 2.5.

## 4 User Study Details

### 4.1 IRB Approval, Participant Sourcing, and Compensation

Our user study, which received approval from our institution's IRB, was conducted using the Amazon Mechanical Turk (MTurk) crowdsourcing platform. Our surveys were made available to MTurk workers with at least 1000 completed HITs (MTurk tasks) and a HIT approval rate of at least 95%. Workers accepted a consent statement (reproduced below) in order to proceed, including confirmation of being age 18 or above. Workers were fully anonymized other than collecting their MTurk worker IDs, a unique non-identifiable code associated with each worker. Workers were compensated according to the length of each survey: the shorter 15-minute survey (version A; *kiki-bouba*) paid \$2.50 upon completion and the longer 25-minute survey (version B; random pseudowords) paid \$4.25, as seen when accepting the task.

### 4.2 Consent Statement

To participate in the survey, workers were required to read and agree to the following consent statement (contact information has been redacted but was visible to workers; the listed time depended on the survey version):

*Consent to Participate in Online Survey Research Using MTurk*

*Study Description: We are researchers at the Tel Aviv University doing a research study about cognition, vision and language. If you agree to participate, you will be asked to complete an online survey that will take approximately (time) minutes to complete.*

*Risks/Benefits: Risks to participants are considered minimal. Collection of data and survey responses using the internet involves the same risks that a person would encounter in everyday use of the internet, such as fatigue or breach of confidentiality. While the researchers have taken every reasonable step to protect your confidentiality, there is always the possibility of interception or hacking of the data by third parties that is not under the control of the research team. There will be no costs for participating. Benefits of participating include payment from Amazon as described in the HIT description.*

*Confidentiality: Researchers will have access to your MTurk worker ID which may be able to link to your personal information on your Amazon public profile page, depending on your settings you have on your Amazon profile. MTurk worker IDs will not be shared with anyone outside the study team and will be used solely for the purposes of distributing compensation and will not be stored with your responses. We will not be accessing any personally identifying information about you that you may have put on your Amazon public profile page. Any reports and presentations about the findings from this study will not include your name or any other information that could identify you.*

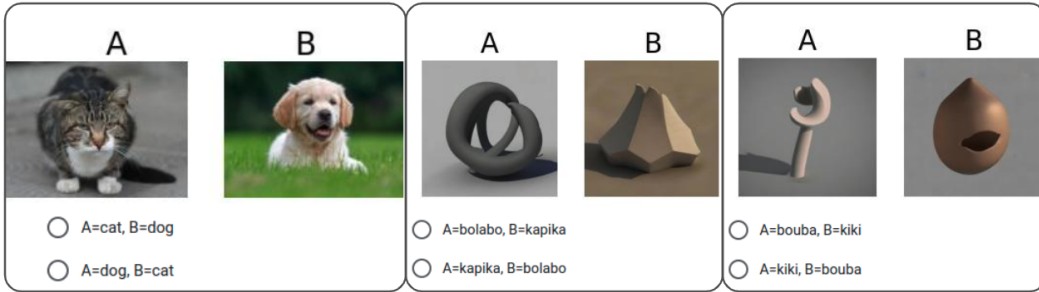

Figure 2: **Screenshots of questions from our user study.** The left-hand screenshot shows the sample question shown to all participants at the beginning of the survey. The screenshot in the center shows a question from the pseudowords version of the survey, while the right-hand screenshot shows a question from the *kiki–bouba* version.

*Voluntary Participation: Your participation in this study is voluntary. You may choose to not answer any of the questions or withdraw from this study at any time without penalty. Your decision will not change any present or future relationship with the Tel Aviv University or Amazon.*

*For more information about the study or study procedures, contact (redacted).*

*Research Subject's Consent to Participate in Research: By entering this survey/ selecting I agree, you are indicating that you have read the consent form, you are age 18 or older and that you voluntarily agree to participate in this online research study. Please make sure that you have read and agree to Amazon's Mechanical Turk participant and privacy agreements as these may impact the disclosure and use of your personal information.*

## 4.3 Survey Instructions

Participants in our user study were shown the following instructions describing the task to complete:

*In the following section you will be shown pairs of images, labelled as "A" and "B" and showing two objects. For each image pair, you will be asked which object is better described by different made-up words. Choose whichever feels more appropriate for the given images, even if you are not sure. For example, in the question below you should pick "A=cat, B=dog".*

Underneath this appeared the sample image pair and options shown on the left of Figure 2. Upon proceeding, they were given the full page of 20 questions beginning with the following instruction:

*You will now see 20 image pairs and will be asked which nonsense words best match each image. Pick whichever feels most right to you, even if you are not sure.*

The 20 questions following this were in the format illustrated in Figure 2, either as in the center or the right-hand side of the figure depending on the survey version. The order that questions appeared was fully randomized for each participant. All of these questions were forced-choice, and participants could only submit after answering all of them.

Finally, participants were asked the (required) question

*Had you heard of the "kiki-bouba effect" before taking this survey?*

along with an optional field for comments.

## 4.4 Survey Design Methodology

We distributed two primary versions of our survey separately: version A, using images generated with prompts using *kiki* or *bouba* (and asking participants to choose between these words); and version B, using images generated with pseudowords from $\Psi_{\star}$ and $\Psi_{\bigcirc}$ (and asking participants to choose between them). Version B was distributed in two variants (B1 and B2) with different (and completely disjoint) sets of pseudowords and image generations.

For all surveys, we selected images by using the image closest to the centroid in CLIP embedding space among images in a 50-item minibatch of generations (as described in Section 2.1), in order to reduce variance due to the stochastic nature of the image generation process.

In version B of the survey, each question displayed one image generated by a pseudoword from $\Psi_{\star}$ and another generated by a pseudoword from $\Psi_{\bigcirc}$, as seen in the right-hand screenshot in Figure 2. No pseudoword appeared in more than one question. We randomly selected pseudowords for use in these questions, but in order to focus on the presence of sound symbolism and avoid participants easily identifying real objects, we manually removed pseudowords close to real English words, as discussed in Section 3.7.

## 4.5   Results Analysis

175 people participated in our survey: 100 saw version A, 50 saw version B1, and 25 saw version B2. Of these, 87 reported that they had not heard of the *kiki–bouba* effect before taking the survey (47 for A, 30 for B1, 10 for B2); we take this into account below and show that our results have similar implications whether including these participants or not.

Overall, participants answered correctly more often than the 50% baseline accuracy expected from random guessing: They answered with 73% accuracy in the *kiki–bouba* setting (version A) and 55% in the general pseudoword setting (versions B1 and B2). When only considering participants who reported that they had not heard of the effect before, the accuracy values were 70% and 56% respectively.

To determine statistical significance and the overall effect size, we control for inter-subject and inter-item variation with a mixed-effects logistic regression model, using the `Lmer` implementation in the `pymer4` library [4]. We regress whether a question is answered correctly by a respondent (categorical response variable: 0 for incorrect answer, 1 for correct answer), treating question identity as a fixed effect and respondent identity as a random effect. We perform this separately for setting A and setting B; in the former, each subject answered all 20 questions; in the latter, each subject answered either the 20 questions of version B1 or the 20 questions of version B2, 40 distinct possible question categories total.

For setting A (*kiki* vs. *bouba*), our model results in an intercept estimate corresponding to a 89% overall success probability ($p < 0.001$) when calculated over all 100 respondents. When using only the 47 who reported not having heard of the effect, the corresponding intercept estimate is 86% ($p < 0.001$). For setting B (random pseudowords), our model results in an intercept estimate corresponding to a 78% overall success probability ($p < 0.001$) when calculated over all 75 respondents. When using only the 40 who reported not having heard of the effect, the corresponding intercept estimate is 74% ($p < 0.015$).

Overall, we find that the sound symbolism exhibited by our text-to-image model significantly accords with human sound symbolic associations; these results are reflected whether or not we restrict our analysis to subjects who had not heard of the *kiki–bouba* effect before.