# OpenReview forum: "Kiki or Bouba? Sound Symbolism in Vision-and-Language Models"
_NeurIPS.cc/2023/Conference — NeurIPS 2023 spotlight_

### Official Review · Reviewer_DdD9 · 2023-07-03

**Soundness:** 3 good
**Presentation:** 3 good
**Contribution:** 2 fair
**Rating:** 6
**Confidence:** 3

**Summary:**

The Kiki-bouba effect is a well studied phenomena in humans to consistently associate sharp and smooth objects with certain phonetics. This paper explores whether such an effect is present within image and language models. Specifically it looks at Stable Diffusion (a text to image generative model) and CLIP (a discriminative model for text and images) to see whether these models form representations of made up "Kiki" and "Bouba" type words in similarly consistent ways as humans do.
The paper is well written, and concludes that these models do indeed demonstrate this same tendency.

**Strengths:**

The paper is well written. The methods used to investigate whether this effect is present in these different models appear valid.
The observed effect is strong. It's interesting that these models demonstrate this as strongly as they do given they have no auditory dimension, but then again they are models of language. Nice observation though!

**Weaknesses:**

It would have been nice to make use of a model that doesn't have have any exposure to images. For example taking BERT and seeing if the pseudo words also get separated with the same tendency or not? Something analogous to demonstrating kiki-bouba in blind people, which to my lay understanding I believe has been done but only to a limited extent compared to other validations of the effect in various human populations.
I'm unsure if this paper opens up any further significant study, or unlocks anything in an engineering sense, so I think its impact is likely capped.
Minor: citation on line 174 would be good, rather than just "some prior works".
Nitpick: Line 24 -- "bœuf" is often translated as "beef" I thought, in which case there is a lot of overlap in sounds. This link is no doubt cultural, rather than saying anything about objects having similar sounding terms across languages. Maybe there is a better example to use though?

**Questions:**

Did you try other ways of scoring? Does distance from centroids of the sharp/smooth groups also display the same effect for example? How important to observing this effect are the details in how scoring is done?

**Limitations:**

The authors do address the fact that these models don't have the same full set of inputs as humans do. They also note that nothing is discussed about why these models display these traits. These are important to highlight, but the paper has value as an observation that the effect is present.

---

> ### Author Rebuttal · Authors · 2023-08-08
>
> We thank the reviewer for the constructive feedback and comments. We will clarify wording issues and add requested citations in a revised version.
>
> Regarding unimodal (text-only) models, we refer to our investigation of these models in the supplementary material (supp Sec 3.1), where we test encoder-only models (similar to BERT) that have been trained on text alone. Our findings show mixed results, with possible sound symbolic effects evident but seemingly weaker than those shown by multimodal models; we leave a thorough investigation of the relative contributions of text and image data to future work (supp L124).
>
> Regarding scoring methods –  see the response to reviewer kkX7 for an additional zero-shot scoring method. Regarding the question about cluster centroids for scoring, our phonetic scoring method measures distance along the axis which separates the two centroids of pseudowords clusters, effectively comparing relative distances to the centroids of each cluster.
> Regarding the effect of different scoring methods, we found sound symbolic effects across probing methods (geometric and phonetic scoring, and the additional method proposed in our response to reviewer kkX7). We also found scoring to be robust to the choice of prompt text used in probes (supp Sec 3.2).
>
> Regarding the French word “boeuf”, we clarify that this is being used on L23 as a direct citation from Saussure, who used this example verbatim in the source being cited. Additionally, “beef” is actually a loanword in English from Old French [1], making the similarity non-coincidental. Nevertheless, we can replace this with an unambiguous example to avoid confusion.
>
> [1] D. Harper. “Etymology of beef.” Online Etymology Dictionary.

---

### Official Review · Reviewer_kkX7 · 2023-07-04

**Soundness:** 3 good
**Presentation:** 4 excellent
**Contribution:** 3 good
**Rating:** 7
**Confidence:** 3

**Summary:**

The work's goal is to study whether sound symbolism is reflected in vision-and-language (VL) models like CLIP and Stable Diffusion (SD). The work proposed a method called **zero-shot knowledge probing**, and verified a sound symbolism phenomenon like **kiki-bouba effect** by evaluating the outputs from CLIP and SD when inputs were a set of predefined pseudowords, adjectives, and nouns, which are related to sharpness or roundness.

In short, the study provides a simple paradigm for investigating sound symbolism in VL models from VL latent space and verifies the existence of sound symbolism in VL models.


**Strengths:**

1. The work gives a clear explanation of motivation, methodology, and experimental results.

2. The work discusses the experimental results thoroughly and with some insights.

3. The work provides rich supplementary material, which answers a few questions raised when I was reading the main paper.

4. The work is well organized and tells a beautiful story with linguistic and cognitive backgrounds.

In conclusion, I believe the work is solid and insightful.


**Weaknesses:**

1. The work only studies open-sourced VL models CLIP and SD, which may relate to narrow computational resources. But studying models with different backbones like GAN or ViT would be more convincing.

2. The **zero-shot knowledge probing** is designed a bit intuitively without comparison with other baseline methods, such as training an individual classifier for text-text and text-image scores or human annotators.

3. The work did not discuss the inner results sound symbolism phenomenon in perspectives of the mechanism of machine learning but attributed it to the inherent knowledge in the models. I think the reason for the phenomenon might be a result of tokenization, i.e. a letter-level tokenized VL model / language model might produce different results.

4. The work only studies the phenomenon in English, given L41 saying the phenomenon is universal to different languages.

**Questions:**

1. Could you give a result on how exactly $|w_{adj}|$ and $|v_{pw}|$ are in L206 and L217 since the distribution of the adjectives and pseudowords in latent space is quite a blur?

2. The caption of Table 2 is not clear, how is the list of words sorted and what does the `POS` line mean?

3. Why do you choose SD to map texts to images, rather than using retrieval methods with CLIP on the LAION-2B dataset? They have the exactly same inherent knowledge.

4. What are the probable applications of this study?


**Limitations:**

Although the work emphasizes that its study should be taken in context (L309), the inner reasons why VL models have the same sound symbolism effect as humans are not investigated. It should be addressed in future works.

---

> ### Author Rebuttal · Authors · 2023-08-08
>
> We thank the reviewer for the constructive feedback and comments. We address the reviewer’s question regarding tokenization in our global response (item C), and the concern about multilingual results in our global response (item B) and in the response to reviewer NTGx.
>
> To address concerns about the generality of our method across different model architectures, we have also tested the SOTA text-to-image models DeepFloyd-IF [1] and Kadinsky [2]. We find significant sound symbolic effects in both models; please see Table 1 in the PDF attached to our global response for full results when evaluating on our pseudowords. For example, DeepFloyd-IF shows geometric score AUC 0.63 and phonetic score AUC 0.98, significantly higher than 0.50 expected from random chance. We note that these models differ in underlying architecture, training objectives, and with respect to training data. For example, DeepFloyd-IF uses a T5 transformer text encoder (rather than the CLIP text backbone used by Stable Diffusion) and uses a pixel-level denoising U-Net (rather than the latent diffusion architecture and objective used by Stable Diffusion). We will include these results and discussion in a revised version.
>
> Regarding probing methods, we focus on zero-shot knowledge probing rather than trained text or image classifiers because we are interested in the inherent knowledge of our models and not the dynamics of training on new data, (L183-184), consistent with prior work on probing models’ intrinsic knowledge [3].  To provide an additional zero-shot method to complement our results, we also analyze image generations using image-level geometric properties. In particular, we estimate the “sharpness” or “roundness” of an image by using a Harris corner detector, which looks for points of sharp discontinuity representing geometric corners. We estimate the number of corners in generated images and find a significant difference between images generated from the two pseudoword classes, with “sharp” pseudoword images having significantly more corners on average than “round” pseudoword images, confirming that the former are visually more sharp than the latter on average. We will include these results in a revised version.
> We first wish to clarify the wording and terminology asked about in questions 1 and 2. $\left|w_{adj}\right|$ as defined on L206 is a unit vector pointing in the direction in CLIP’s latent space which best separates between the two sets of adjectives from L205 (inserted into the prompts from L187). $\left|v_{pw}\right|$ as defined on L217 is similarly a unit vector pointing in the direction in CLIP’s latent space which best separates between the two sets of pseudowords (L176, inserted into the prompts from L187). In Table 2, POS stands for “Part Of Speech” (in our case, noun or adjective) and the words are sorted by phonetic score $\phi_{\left<w\right>}$. We  will include these clarifications in a revised version.
>
> Regarding the use of Stable Diffusion (SD) for text-to-image generation versus image retrieval, we believe that there is in fact an important difference in the intrinsic knowledge of SD versus CLIP-guided image retrieval on LAION. While SD and (Open)CLIP were both trained on LAION, the intrinsic knowledge learned by SD’s denoising goal (which requires understanding local regions within images) is likely to be different from knowledge gained from image-level retrieval. Indeed, our quantitative and qualitative results demonstrate that CLIP and SD do not behave identically with respect to intrinsic knowledge; with SD) showing stronger sound symbolic effects (e.g. L299). Our primary motivation for including SD in our tests is to investigate the knowledge learned by a popular generative model, in addition to probing the discriminative model CLIP directly.
>
> Regarding applications of our work, we see value in understanding how these models “interpret and respond to language” (L335); in general, V&L models are being used as black boxes without a full understanding of how they understand the visual semantics of language, and we believe that probing them to understand what they have learned may be relevant to model interpretability. We also believe that our findings could have applications to cognitive science and linguistics (L339-344), where sound symbolism has long been a topic of interest and debate; computational methods could be used to provide more evidence for the presence of sound symbolism and to understand its cognitive basis. We will make these points more explicit in a revised version.
>
> Finally, we wish to clarify the wording and terminology asked about in questions 1 and 2. $\left|w_{adj}\right|$ as defined on L206 is a unit vector pointing in the direction in CLIP’s latent space which best separates between the two sets of adjectives from L205 (inserted into the prompts from L187). $\left|v_{pw}\right|$ as defined on L217 is similarly a unit vector pointing in the direction in CLIP’s latent space which best separates between the two sets of pseudowords (L176, inserted into the prompts from L187). In Table 2, POS stands for “Part Of Speech” (in our case, noun or adjective) and the words are sorted by phonetic score $\phi_{\left<w\right>}$. We will include these clarifications in a revised version.
>
> [1] DeepFloyd/IF-I-M-v1.0 on Hugging Face Model Hub
>
> [2] kandinsky-community/kandinsky-2-2-decoder on Hugging Face Model Hub
>
> [3] Petroni et al. “Language Models as Knowledge Bases?” EMNLP 2019

---

> > ### Comment · Reviewer_kkX7 · 2023-08-11
> >
> > Thanks for your response.
> >
> > > the concern about multilingual results in our global response (item B)
> >
> > It is a reasonable reply. It will be better if the linguistic feature of different languages is considered in the additional experiments.
> >
> >
> > > We note that these models differ in underlying architecture, training objectives, and with respect to training data.
> >
> > "training objectives": I do not think their training objectives differ a lot. It might still be worth experimenting with GAN-style models.
> >
> > > While SD and (Open)CLIP were both trained on LAION, the intrinsic knowledge learned by SD’s denoising goal (which requires understanding local regions within images) is likely to be different from knowledge gained from image-level retrieval.
> >
> > From this perspective, you should define "intrinsic knowledge" more clearly that it is the knowledge learned by denoising training objective of SD in the revised paper.

---

> > > ### Author Response · Authors · 2023-08-12
> > >
> > > Thank you for your reply and additional comments.
> > >
> > > Regarding linguistic features of languages tested, we will provide additional details, such as the language families of each language tested (Hungarian - Uralic/Ugric, Indonesian - Austronesian/Malayo-Polynesian, Finnish - Uralic/Finnic, Lithuanian - Indo-European/Baltic) and explaining how these differ from one another and from English, in a revised version.
> > >
> > > Regarding GAN-style models, we have also experimented with GALIP [1], a GAN-based text-to-image generation model. Using GALIP pretrained on the CC12M dataset with our pseudoword methodology, we see significant sound symbolic associations via our metrics and qualitatively - for example, phonetic score AUC 0.62 and geometric score AUC 0.98, significantly higher than 0.50 expected by chance for each. We will include these results in a revised version.
> > >
> > > Regarding intrinsic knowledge, we will clarify this wording in a revised version.
> > >
> > > [1] Tao et al. GALIP: Generative Adversarial CLIPs for Text-to-Image Synthesis. CVPR 2023

---

### Official Review · Reviewer_gvCK · 2023-07-06

**Soundness:** 3 good
**Presentation:** 4 excellent
**Contribution:** 4 excellent
**Rating:** 7
**Confidence:** 4

**Summary:**

The authors present an investigation of the phenomenon of ‘sound symbolism’ in a pair of cutting edge Vision-and-Language Models. Sound symbolism is an intriguing phenomenon whereby the meaning of a word can be in part traced back to the way the word sounds. It is an important phenomenon in psychology and linguistics because it challenges a strict view of the mapping between form and meaning as arbitrary (stated most famously by Saussure). The authors probe CLIP and Stable Diffusion by creating a small dataset of pseudowords that were constructed to reflect phonetic features that generally map onto “sharp” vs. “round” speech sound categories. They project the embedding vectors in 1024-dimensional CLIP space onto a one-dimensional semantic dimension of interest, defined by two sets of antonym shape adjectives (one set corresponding to synonyms of ‘round’ and the other corresponding to synonyms of ‘sharp’). They also define a phonetic score to measure phonetic/graphemic associations of the shape adjectives with the pseudowords. They analyze the models using these geometric measures, as well as with a human evaluation where participants are asked to select which of two Stable Diffusion-generated images is best described by a pseudoword. Across these evaluations, they find relatively strong evidence for the presence of sound symbolism in these models.

**Strengths:**

One of the strengths of this paper is that it demonstrates how ideas from Cogntive Science can drive investigations into modern AI models. The phenomenon of sound symbolism is particularly interesting both for its intuitive appeal and its relation to foundational ideas in Cognitive Science. The paper was well written and largely a pleasure to read. The methods were overall well described and I found the evidence relatively convincing.

**Weaknesses:**

The weakest part of the paper for me were the motivations, hypotheses, and implications. Yes, Vision&Language Models are increasingly powerful and increasingly deployed, but why are they interesting objects of study for Cognitive Science? We are told that “these methods could provide new insights into the classic questions of what aspects of sound are tied to meaning”, but not explained how this might happen or what this could look like. As well, the conclusion about “cultural universality” seems like a very difficult and non-obvious question that these methods would provide insight into.

And the flip-side, why is sound symbolism an important phenomenon to investigate in these powerful and deployable models? One specific weakness is that there were no hypotheses presented for the models, and you could imagine developing the hypotheses based on how these models were trained. I think there’s a lot to say about this, but all we got one sentence buried in the results section that mentioned that the models primarily saw valid English text during training and didn’t have access to the actual sound of the words.

The second major weakness for me was the description and analysis of the User Study. The authors write “Over two hundred volunteers participated in this study”. The use of the word “volunteers” here suggests that the participants were not paid, which I hope is not the case. Who are these participants? How were they recruited? Was there any ethical review of this study? How many participants exactly were recruited (this is in the supplement, but should be presented in the main text)?

On the analysis side, the use of a binomial test is not appropriate as the individual data points are not i.i.d. Participants gave multiple responses and, presumably, multiple items (pairs of images) were rated by multiple participants. Thus, something akin to a mixed-effects (multi-level / hierarchical) logistic regression model would be more appropriate to account for the participant- and item-wise variability, and provide a much more realistic analyses of the results (see Gelman & Hill, 2006 for background).

I found the Phonetic Scores in 3.2 the most difficult to understand and still am not completely sure I get it. From the math, I would have thought this score was something akin to a control measurement, and a way to contextualize the Geometric scores, but the authors seem to have a different perspective, which I didn’t fully get.

Finally, Figures 2 & 3 took up a lot of space for not that much information. They could easily be condensed to make room for some of the context in the response.


**Questions:**

My questions are the same as the weaknesses. I’m not sure all of questions 1-3 need to be answered, but at least 2 of them do.
1. How can these results inform theories in Cognitive Science?
2. What hypotheses can be derived from these models a priori about sound symbolism, presumably by thinking about their training data?
3. Why is sound symbolism important for evaluating sophisticated, deployable AI systems?
4. Can you unpack the Phonetic Scores in a little more detail, and explain their relevance?
5. Please provide more details about the User Study and analyze the results in a more appropriate manner.


**Limitations:**

The limitations were okay but could be improved by thinking more about the societal impact of these (human) biases getting incorporated into deployable generative models. I would bet some of the impacts of sound symbolism have been discussing in the psychological literature, and it would be important to mention them and discuss them in the context of these things getting baked into generative AI.

---

> ### Author Rebuttal · Authors · 2023-08-08
>
> We thank the reviewer for the constructive feedback and comments. We address the reviewer’s concerns regarding our user study in our global response (item A).  In particular, we reiterate that we believe the user study to not be critical to our results, and we are willing to remove or replace it upon request.
>
> Regarding motivation, we first note that other reviewers mention the motivation of our work as a strength, referring to it as an “interesting study that is well-motivated” (NTGx), having a “clear explanation of motivation… [and telling] a beautiful story” (kkX7). We explicitly address the reviewer’s questions next, emphasizing aspects of our work which we will better illustrate in a revised version.
>
> Regarding question 1 (importance of our results in the context of cognitive science), we believe our results have an impact in this domain for a number of reasons. Firstly, they suggest that sound symbolism is reflected in the linguistic training data of these models, and the presence of sound symbolism in language itself is moderately controversial (notably denied by Saussure, L22-23). Additionally, a series of psychological and linguistic works have investigated whether sound symbolism is reflected in the basic lexicon of English (see citations on L94); if V&L models learn sound symbolic association from valid English text (or other languages) in caption data, it might be due to this effect and could confirm the results of these studies with a new methodology. Finally, investigation into how V&L models infer sound symbolic associations from their training data could potentially shed light on how sound symbolism is learned during human language acquisition. While we do not claim to directly answer these questions, our results suggest that V&L models could provide valuable insights for investigating them. We will state these points more explicitly in a revised version.
>
> Regarding question 2 (a priori hypotheses regarding models) – when considering text containing nonsense words as input, one might expect models such as Stable Diffusion (SD) to produce purely random or nonsensical output. The competing hypothesis, which we believe to be non-obvious but for which we find strong evidence, is that such models have learned associations between the individual characters used in these nonsense words, and moreover in accordance with associations known from the psycholinguistic literature.
>
> Regarding question 3 (importance of our results in evaluating AI systems) as well as the reviewer’s comment about limitations and bias in generative models – in general, we believe that it is important to investigate what is learned by these powerful and deployable models and to provide interpretability to their behavior. In particular, we see value in understanding how V&L models may learn patterns from their training data, particularly patterns which require complex generalization beyond the captions seen during training. While we do not claim to determine the source of the observed effects, we provide evidence that these models have not simply memorized specific pseudowords seen during training (Sec 4.5). Such generalization could shed light on mechanisms leading to various associations and biases exhibited by V&L models, suggesting that they cannot only be explained by looking at individual training instances but possibly requiring an understanding of generalization over complex patterns on a large scale. We will incorporate such a discussion in a revised version.
>
> Regarding the statistical analysis of our survey results, we thank the reviewer for the suggestion and re-analyze the data using a mixed-effects logistic regression model. In this setting, we regress whether a question is answered correctly by a respondent (categorical response variable: 0 for incorrect answer, 1 for correct answer), where the question identity corresponds to a fixed effect and the respondent identity corresponds to a random effect. We fit this model using the “Lmer” implementation in the lme4 R package for modeling of linear mixed-effects models. For example, analyzing our survey of “kiki / bouba” generations* results in an intercept estimate corresponding to 89% overall success probability (p≈0.001), which isolates the overall success rate from the effects of individual respondents and questions. We will use this statistical analysis in a revised version (contingent on use of our survey data, per the discussion above).
>
> *(treating questions from survey versions 1 and 2 as separate question categories, and only using data from respondents who had not previously heard of the kiki-bouba effect)
>
> Regarding the phonetic scoring method (Sec 3.2), we clarify that the probe vector (L217) measures the direction in CLIP’s embedding space between the centroids of the two pseudoword classes (either as text or image embeddings, depending on the model being tested). We call this “phonetic scoring” because it is a semantic dimension determined by the phonetics (sound classes) of pseudowords alone. In Table 1 (last two columns) and Figure 3 we examine correlation between ground-truth adjective class and position on this dimension. In Table 2, we show that this dimension in CLIP space corresponds to interpretable semantics of real English words. We will emphasize these clarifications in a revised version.

---

> > ### Comment · Reviewer_gvCK · 2023-08-16
> > **Thank you**
> >
> > I appreciate the very thoughtful rebuttal. Your explanation surrounding the motivation for the study (i.e., importance in the context of CogSci and a priori hypotheses about models) is well presented and thought provoking. I think this was the key thing missing for me with respect to motivation, and you’ve adequately addressed my concern here. So please put this in the paper.
> >
> > I also like your comments regarding the importance in evaluating AI systems with this perspective. So please include that in the paper as well.
> >
> > I also very much appreciate you implementing a more principled statistical analysis, so well done.
> >
> > Finally, regarding the user study: I don’t have strong feelings about whether to include or not (but you may want to consult the Ethical guidelines at NeurIPS about its inclusion). What I do feel strongly about is that you describe the protocol with more precise language (as you did in the rebuttal). Eg., in the paper you should write the population of participants you recruited (i.e., university graduate students) and that they were unpaid volunteers. Also, reporting the precise number that you collected in the main text, etc.

---

> > > ### Author Response · Authors · 2023-08-17
> > >
> > > Thank you for your response. We will include all of these points in our revised paper.

---

### Official Review · Reviewer_NTGx · 2023-07-07

**Soundness:** 4 excellent
**Presentation:** 4 excellent
**Contribution:** 3 good
**Rating:** 8
**Confidence:** 4

**Summary:**

This paper determines the extent to which pretrained vision and language models encode phonetic information associated with sharp or round objects. Prior research has shown a cross-lingual tendency to associate sounds with shapes in human studies. In this work, the authors investigate whether this holds for a discriminative (OpenCLIP) or generative (StableDiffusion) machine learning model. The methodology starts by constructing a set of 648 pseudowords based on a set of sharp / round Latin letters. Two prompts are used with the models, which either have the pseudoword used as a noun or an adjective. The prompts and pseudowords are used to show that both models do indeed exhibit a similar effect to humans. A human study shows a much stronger effect for the original kibi-bouba pair than the 648 pseudowords. The paper also claims that this finding cannot be attributed to the models seeing these examples during pretraining.

**Strengths:**

* Interesting study that is well-motivated.
* Claims are clearly supported in the experiments.
* Human evaluation is used to further support the automatic evaluation metrics.

**Weaknesses:**

* No major weaknesses that I could identify

**Questions:**

* Do these results hold with a multilingual CLIP model or in a multilingual human evaluation?

**Limitations:**

* The paper does a good job of discussing potential limitations of the research

---

> ### Author Rebuttal · Authors · 2023-08-08
>
> We thank the reviewer for the constructive feedback and comments.
>
> To investigate whether sound symbolism may exist in a multilingual V&L model, we test the Kadinsky [1] multilingual text-to-image model (which uses multilingual CLIP) with our methodology, on four geographically and linguistically diverse languages: Finnish, Indonesian, Hungarian, and Lithuanian. Please see Tables 2 and 3 in the PDF attached to our global response for full results including metrics and prompt texts in each language. We find non-trivial sound symbolism in this setting in each language; for example, Finnish displays geometric and phonetic AUC 0.69 and 0.94 respectively, significantly higher than the 0.50 expected from random chance. These results suggest that sound symbolism may be learned in a multilingual V&L setting.
>
> Even with the results of this additional experiment, we wish to emphasize that we do not claim to demonstrate the universality of sound symbolism (as suggested by reviewer Cwqz). Although this is a topic of interest in the psychological literature, our work focuses on showing the existence of this phenomenon in V&L models, and not on whether it is a universal phenomenon cross-linguistically. We will state this more explicitly in a revised version of our work.
>
> [1] kandinsky-community/kandinsky-2-2-decoder on Hugging Face Model Hub

---

> > ### Comment · Reviewer_NTGx · 2023-08-14
> >
> > Thank you for running an additional experiment using a multilingual model. The results are interesting and it makes me more confident in my assessment of the work.

---

### Author Rebuttal · Authors · 2023-08-08

We thank the reviewers for their constructive comments. We respond here to shared concerns as well as items raised by reviewers JJeH and Cwqz, referring to individual responses where more information is given.

**(A) Survey – reviewers JJeH, Cwqz, gvCK**

The main focus of our work is automatic probing of computational models for sound symbolism; the user study suggests that these results are grounded in human cognition, but it does not test our central hypothesis (whether V&L models have learned sound symbolism). We saw value in showing this additional result and conducted the survey following the informal procedure described below, but we are willing to either remove it entirely or to replace it with a study using Mechanical Turk crowdworkers (following all relevant procedures) upon request.

Regarding the methodology used, we used volunteers to avoid concerns about crowdsourcing and contract work, under the good-faith interpretation of NeurIPS’s ethics code that human studies refer to paid workers rather than consenting volunteers. This interpretation was made following discussions with several peers in our institutions, as well as observing that this practice has been followed in papers published in NeurIPS 2022 such as:

* Zhang et al. Generalized One-shot Domain Adaptation of Generative Adversarial Networks. (“...we finally collect valid votes from 53 volunteers…”)
* Hu et al. Hand-Object Interaction Image Generation. (“...we conduct a user study... There are 20 volunteers participating in this study.”)

Following common practices in our institutions, our study was distributed among university graduate students who were not familiar with this research project, and we controlled for prior knowledge of the phenomenon (supp L196, L227). We controlled for native language (supp Table 5) as this would be a natural confounding factor. We did not collect any sensitive personal information or demographic data such as race or education level. While reviewer Cwqz mentions these as significant for showing the universality of sound symbolism, we clarify that this is not the stated aim of our work (noting that L32-35 are not discussing our research goals, but rather providing context from the psychological literature). We will clarify these points in a revised version (contingent on any decision regarding the inclusion of the user study).

**(B) Multilingual evaluation – reviewers Cwqz, NTGx, kkX7**

Please see the response to reviewer NTGx where we conduct an additional experiment to test the cross-lingual generality of our results. Full results for each language tested are shown in Table 2 of the attached PDF, and Table 3 shows the prompts used for each language. We find evidence for the effect in a multilingual V&L model for each of the four geographically and linguistically diverse languages, and we will include these results in a revised version.

**(C) Tokenization – reviewers Cwqz, kkX7**

Regarding tokenization, we agree that this could potentially contribute to the observed effect. However, we wish to emphasize that our approach tests V&L models end-to-end, agnostic to the source of these effects with respect to the relative contribution of different model-internal components. We do note that OpenCLIP and Stable Diffusion use the same tokenizer and yet show differences in the strength of their sound symbolic effects (e.g. L299), precluding this as the only source of the observed results. We also clarify that we do not adjust the vocabulary of the models or the tokenizers in any way; we use these models as-is and probe them in the zero-shot regime. We will further emphasize these clarifications in a revised version.

Regarding reviewer Cwqz’s concern about items being “correctly mapped to their corresponding word vectors in CLIP”, we clarify that we deliberately design our pseudowords to avoid valid English words (L173) and thus they are indeed not found in the tokenizer’s vocabulary. This does not mean that the models are incapable of dealing with these inputs. This can be observed with many common English words, which are also not found in its vocabulary. For instance, “handkerchief” is split into three subword tokens (“hand”+”ker”+”chief”) by the tokenizer, but Stable Diffusion generations for this word show that it does indeed understand what a handkerchief is as a concept. Text encoders are capable of processing words that are not found in their vocabulary by splitting them into subword tokens, and we leverage this property to probe sound symbolism in these models.

**(D) Pseudoword construction – reviewer JJeH**

We clarify that the construction of pseudowords was fully automatic with no involvement of human participants. These were constructed using the combinatorial procedure described on L167-168 to include all possible combinations of letters matching the given pattern.

**(E) Vision-audio models – reviewer Cwqz**

While we agree that investigating such models would be an interesting and promising line of research, we base our investigation on an abundance of prior works in psychology and linguistics which investigate sound symbolism in written language. There is an implicit mapping between text and sound in spoken and written language, and the term “sound symbolism” is frequently used to refer to the possible connection between graphemes in text and meaning, as stated explicitly by [1, 2, 3] (which we cite on L88-91). As such, studying models trained on acoustic data would be an interesting direction for further research, but is out of scope of our study. We will emphasize this point in a revised version.

[1] Cuskley et al. “Phonological and orthographic influences in the 384 bouba–kiki effect”

[2] De Carolis et al. “Assessing sound symbolism: Investigating phonetic forms, visual shapes and letter 375 fonts in an implicit bouba-kiki experimental paradigm”

[3] Cwiek et al. “The bouba/kiki effect is robust across cultures and writing systems”

---

### Decision · Program_Chairs · 2023-09-21

**Decision:**

Accept (spotlight)

**Comment:**

This paper examines the extent to which vision-and-language models capture sound symbolism - particularly, the "kiki" vs. "bouba" effect, whether models encode a relation between phonetic sounds and sharp or round shapes. They study the OpenCLIP and Stable Diffusion models using a set of pseudoword prompts and probing techniques, and find a significant effect in both models. They also conduct a user study to validate that the pseudowords have a similar effect on humans.

Reviewers agree that this is a strong paper, with interesting, well-designed experiments and claims that are clearly supported by the data. There were some questions about motivation for this area of study, which the authors elaborate on in their response, and about multilingual and unimodal (text-only) experiments, which they include in their supplemental PDF. Reviewers also ask about testing a more diverse set of models, since OpenCLIP and Stable Diffusion are closely related.

The paper was flagged for ethics review, however, with some concerns about whether the authors followed appropriate protocols around consent, data, etc. for the user study. The authors state that the participants were volunteers; however, we note that this still counts as a human study and there should still be a formal agreement by which subject consent to participation. We ask the authors to please include in their camera-ready version information on consent and how any personal data was handled, in accordance with the ethical guidelines at https://neurips.cc/public/EthicsGuidelines.